# The SUMO ligase MMS21 profoundly influences maize development through its impact on genome activity and stability

Junya Zhang[1], Robert C. Augustine[1,¤], Masaharu Suzuki[2], Juanjuan Feng[1,3], Si Nian Char[4], Bing Yang[4,5], Donald R. McCarty[2], Richard D. Vierstra[1]*

**1** Department of Biology, Washington University in St, Louis, St. Louis, Missouri, United States of America,
**2** Department of Horticultural Sciences, University of Florida, Gainesville, Florida, United States of America,
**3** State Key Laboratory of Cotton Biology, School of Life Sciences, Henan University, Kaifeng, Henan, China,
**4** Division of Plant Sciences, Bond Life Sciences Center, University of Missouri, Columbia, Missouri, United
States of America, **5** Donald Danforth Plant Science Center, St. Louis, Missouri, United States of America

☯ These authors contributed equally to this work.
¤ Current address: Department of Biology, Amherst College, Amherst, Massachusetts, United States of
America
* rdvierstra@wustl.edu

**Data Availability Statement:** The raw RNA-seq files are available at the NCBI Sequence Read Archive database under the submission number PRJNA685214 (https://www.ncbi.nlm.nih.gov/sra/

## Abstract

The post-translational addition of SUMO plays essential roles in numerous eukaryotic processes including cell division, transcription, chromatin organization, DNA repair, and stress defense through its selective conjugation to numerous targets. One prominent plant SUMO ligase is METHYL METHANESULFONATE-SENSITIVE (MMS)-21/HIGH-PLOIDY (HPY)-2/NON-SMC-ELEMENT (NSE)-2, which has been connected genetically to development and endoreduplication. Here, we describe the potential functions of MMS21 through a collection of UniformMu and CRISPR/Cas9 mutants in maize (*Zea mays*) that display either seed lethality or substantially compromised pollen germination and seed/vegetative development. RNA-seq analyses of leaves, embryos, and endosperm from *mms21* plants revealed a substantial dysregulation of the maize transcriptome, including the ectopic expression of seed storage protein mRNAs in leaves and altered accumulation of mRNAs associated with DNA repair and chromatin dynamics. Interaction studies demonstrated that MMS21 associates in the nucleus with the NSE4 and STRUCTURAL MAINTENANCE OF CHROMOSOMES (SMC)-5 components of the chromatin organizer SMC5/6 complex, with *in vitro* assays confirming that MMS21 will SUMOylate SMC5. Comet assays measuring genome integrity, sensitivity to DNA-damaging agents, and protein versus mRNA abundance comparisons implicated MMS21 in chromatin stability and transcriptional controls on proteome balance. Taken together, we propose that MMS21-directed SUMOylation of the SMC5/6 complex and other targets enables proper gene expression by influencing chromatin structure.

PRJNA685214). The .raw, .msf, .mzid and .mzML
files for the MS datasets are available in the
ProteomeXchange database under accession
number PXD026853 within the PRIDE repository
(http://www.proteomexchange.org/).

**Funding:** This work was supported by grants from
the NSF-Plant Genome Research Program to RDV
(IOS-1546862) and to DRM and MS (IOS-
1748105) https://nsf.gov/funding/pgm_summ.jsp?
pims_id=5338, and a NIH NRSA Ruth L.
Kirschstein postdoctoral fellowship (5 F32
GM103161) to RCA https://researchtraining.nih.
gov/programs/fellowships/f32. The funders had no
role in study design, data collection and analysis,
decision to publish, or preparation of the
manuscript.

**Competing interests:** The authors have declared
that no competing interests exist.

## Author summary

The post-translational addition of SUMO to other proteins by the MMS21 SUMO ligase
has been implicated in a plethora of biological processes in plants but the identit(ies) of its
targets and the biological consequences of their modification remain poorly resolved.
Here, we address this issue by characterizing a collection of maize *mms21* mutants using
genetic, biochemical, transcriptomic and proteomic approaches. Our results revealed that
*mms21* mutations substantially compromise pollen germination and seed/vegetative
development, dysregulate the maize transcriptome, including the ectopic expression of
seed storage protein mRNAs in leaves, increase DNA damage, and alter the proteome/
transcriptome balance. Interaction studies showed that MMS21 associates in the nucleus
with the NON-SMC-ELEMENT (NSE)-4 and STRUCTURAL MAINTENANCE OF
CHROMOSOMES (SMC)-5 components of the chromatin organizer SMC5/6 complex
responsible for DNA-damage repair and chromatin accessibility. Our data demonstrate
that MMS21 is crucial for plant development likely through its maintenance of DNA
repair, balanced transcription, and genome stability.

## Introduction

Plants like other cellular organisms exploit a plethora of post-translational modifications to
expand the functionality of their proteomes, including controls on enzymatic activity, subcel-
lular location, interaction with other effectors, and ultimately on the turnover rates of the
affected proteins. One reversible modification that is emerging as a key regulator involves
attachment of the ~100-amino-acid protein SMALL UBIQUITIN-LIKE MODIFIER (SUMO),
which is structurally related to ubiquitin and likewise becomes covalently linked via an isopep-
tide bond to accessible lysines within its targets [1,2].

Over the past decade, in-depth proteomics have identified over a thousand SUMO sub-
strates in the eudicot *Arabidopsis thaliana* [3–6], with companion genetic studies providing
links between SUMOylation and a wide array of cellular processes. Included are regulations
of gamete formation and embryogenesis, leaf development, root stem cell maintenance,
hormone signaling, light perception, circadian rhythm entrainment, phosphate acquisition,
transcriptional and epigenetic regulation, DNA damage repair, and defense against various
abiotic and biotic challenges [2,7–19]. Particularly notable is the rapid SUMOylation of
numerous proteins when plants are subjected to pathogen attack and heat, drought or salt
stress, which presumably provides protection by yet to be fully understood mechanism(s)
[14,20–24]. Considering that SUMOylation regulates physiological and developmental pro-
cesses crucial to agriculture, uncovering the molecular mechanisms underpinning selective
SUMOylation might reveal novel strategies for crop improvement, especially in suboptimal
environments.

SUMOylation is driven by an ATP-dependent enzymatic cascade involving the sequential
action of a SUMO-activating enzyme (or E1) and a SUMO-conjugating enzyme (or E2) that
prepares SUMO for addition, and in most cases, a SUMO-protein ligase (E3) that identifies
appropriate substrates and encourages transfer from a thioester-linked E2-SUMO donor [1,2].
While some substrates become modified with a single SUMO, others become iteratively modi-
fied with multiple SUMOs attached either at multiple lysines within the substrate or to previ-
ously bound SUMOs connected internally through SUMO-SUMO isopeptide linkages. The
conjugated SUMOs can also be ubiquitylated by a family of SUMO-targeted ubiquitin ligases,

thus merging the influence of these two ligation systems. SUMO attachment is often reversible through a collection of deSUMOylating enzymes that specifically cleave the isopeptide bond between the SUMO moiety and target lysines [25].

To date, three classes of SUMO E3s have been characterized in plants: SAP AND MIZ1 DOMAIN-CONTAINING LIGASE (SIZ)-1, PROTEIN INHIBITOR OF ACTIVATED STAT-LIKE (PIAL)-1/2, and METHYL METHANESULFONATE (MMS)-21/HIGH-PLOIDY (HPY)-2/NON-SMC-ELEMENT (NSE)-2, referred to here as MMS21 [1,2]. They share a SP-RING domain that binds the E2-SUMO intermediate [26], along with a variety of other motifs that presumably identify specific substrates and/or help anchor the E3 to appropriate surfaces/complexes, including DNA and methylated histones. While SUMOylation by the SIZ1 E3 has been connected to a wide range of cellular events and substrates in plants, especially those related to stress defense [5,14,21,23,27], the function(s) of the other E3s are currently unclear.

MMS21, in particular, has garnered interest given its high conservation among eukaryotes and its potential action in a variety of nuclear processes. Arabidopsis *mms21* null mutants are viable but develop stunted roots, altered apical meristems, dwarfed rosettes, and higher chromosome ploidy numbers in both somatic tissue and male gametes, suggestive of roles in the cell cycle and endoreduplication [12,28,29], while the null *mms21* mutants in rice also have stunted vegetative growth [19]. More recently, MMS21 was linked to the DNA damage response and chromatin structure potentially through the respective modification of BRAHMA, a conserved ATPase component within the SWI/SNF chromatin-remodeling complex, and the cell-cycle check point protein DPa, [18,30,31]. In other organisms, MMS21 has been shown to be a crucial component of the nuclear STRUCTURAL MAINTENANCE OF CHROMOSOMES-5/6 (SMC5/6) complex, which is an evolutionarily-conserved ATPase that influences chromatin compaction and is required for recombinational DNA repair, replication fork restart, ribosomal DNA and telomere maintenance, and genome stability [32]. In accord, MMS21 has been found associated with Arabidopsis SMC5/6 complex [13], with one possible target being the integral NSE4 subunit [5].

Here, we further investigated the functions of MMS21 in maize (*Zea mays*) using a library of UniformMu transposon-insertion and CLUSTERED REGULARLY INTERSPACED SHORT PALINDROMIC REPEATS (CRISPR)/Cas9-induced mutations. These analyses revealed that MMS21 is essential in maize with critical roles in root, shoot, pollen, and seed development. While the exact mechanism(s) are not yet clear, defects in DNA repair, and mis-regulated proteome/transcriptome balance were evident for the *mms21* germplasm. Interaction studies, RNA-seq analyses, and *in vitro* SUMOylation assays collectively linked MMS21 to the NSE4 and SMC5 subunits of the SMC5/6 complex, suggesting that MMS21-directed SUMOylation of the SMC5/6 complex and possibly other targets are essential for proper chromatin function and subsequent maize development.

## Results

### The *Mms21* locus is essential for normal maize development

From tBLAST scans of the maize genome using Arabidopsis MMS21 (HPY2) as the query, we identified a single maize *Mms21* locus within the B73 background (GRMZM2G022065), which is located on the long arm of chromosome 6. It spans 4.3 kbp over 7 exons (Fig 1A), and encodes a 245-amino-acid protein with 49% sequence identity to that from Arabidopsis, and 66% and 90% identity to those from the more closely related species, rice and sorghum, respectively. Sequence alignment of maize MMS21 with its plant, yeast, and human orthologs revealed substantial homology throughout the protein, especially within the SP-RING domain

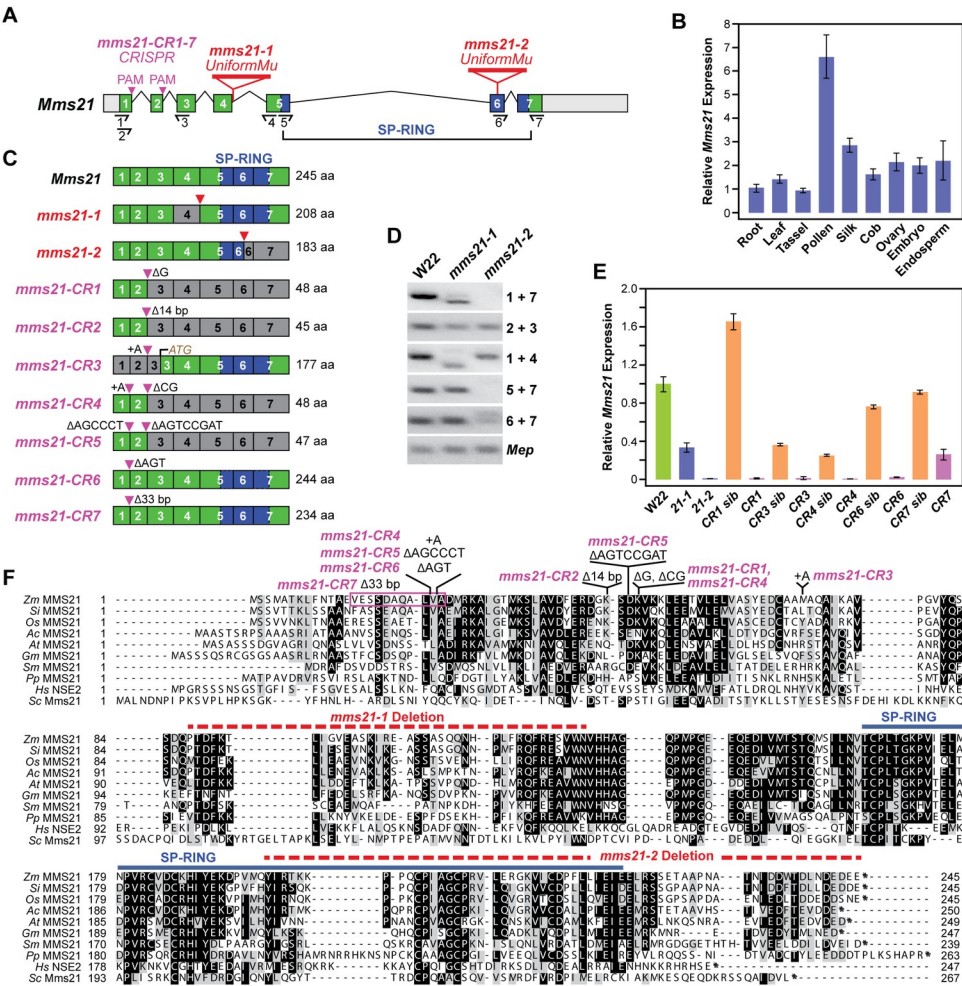

**Fig 1. Description of Maize *Mms21* and a Collection of Mutant Alleles.** (**A**) Gene diagram of *Mms21*. Grey and colored boxes represent UTR and coding regions, respectively. Introns are denoted by bent lines. The coding region for the SP-RING domain is shown in blue. Primers used for RT-PCR analysis in panel D and qRT-PCR analyses in panels B and E are located the half arrows. Locations of UniformMu (*mms21-1* and *mms21-2*) and CRISPR/Cas9 mutagenic sites (*mms21-CR1 to -CR7*) are indicated. (**B**) qRT-PCR analyses of *Mms21* gene expression in different maize tissues. Tissues were sampled from a W22 plant at flowering. The expression level of *Mms21* in roots was arbitrarily set as 1 using expression from *Act1* as an internal standard (n = 4 technical replicates, ±SD). (**C**) *Mms21* mRNA architecture and expected size of the wild-type and predicted mutant polypeptides. Coding region(s) omitted from the mutant *mms21* transcripts are colored in grey. The specific nucleotide insertions and/or deletions in the *mms21-CR* alleles are shown. The predicted ectopic ATG start codon in the *mms21-CR3* allele is indicated. (**D**) RT-PCR analysis of the UniformMu *mms21-1* and *mms21-2* alleles. RT-PCR of the *Mep* locus was used to verify analysis of equal amounts of RNA. (**E**) Comparison of *Mms21* gene expression among the collection of *mms21* alleles. Total RNA was isolated from shoots collected 10-DAS. The expression level of *Mms21* in W22 was arbitrarily set as 1 using expression of *Act1* as an internal standard (n = 4 technical replicates, ±SD). (**F**) Amino acid sequence alignment of MMS21 proteins from plants, yeast and humans, along with the position of the mutations found in the maize allele collection. Black and grey shading indicate identical and similar amino acids, respectively. The amino acid length of the wild-type proteins is shown at the end of the sequence. The region encompassing the SP-RING domain is located by the blue line. The coding sequences expected to be absent in the *mms21-1* and *mms21-2* mutants are located by the dashed red lines. *Zm*, *Zea mays*; *Si*, *Setaria italica*, *Os*, *Oryza sativa*; *Ac*, *Aquilegia coerulea*; *At*, *Arabidopsis thaliana*, *Gm*, *Glycine max*; *Sm*, *Selaginella mollendorffii*; *Pp*, *Physcomitrella patens*; *Hs*, *Homo sapiens*; and *Sc. Saccharomyces cerevisiae*.

that binds the E2-SUMO donor [26]. Quantitative reverse-transcribed (qRT)-PCR showed that the *Mms21* gene is widely expressed throughout maize, including silk, ovaries, and developing embryos and endosperm, with its highest mRNA levels found in pollen (Fig 1B).

Our efforts to characterize *Mms21* genetically began with a search for compromising *Mutator* (*Mu*) insertion alleles within the UniformMu population generated with the W22 inbred [33], which led to the identification of the *mms21-1* allele (mu1068022). Based on genomic PCR and DNA sequence analyses of RT-PCR products, the *Mu* element inserted at the front of intron 4, which induced mis-splicing of the *Mms21* transcript and ultimately fusion of the 3rd and 5th exons (Fig 1C, 1D and 1F). If translated, this mutation should eliminate 37 amino acids upstream of the SP-RING region. Searches of the UniformMu population for seed defects (see below), based on bulk segregation analysis by the Mu-seq method [34], identified a second *mms21* allele—*mms21-2*. Its *Mu* insertion site was confirmed by sequencing PCR products generated with TIR6 and *Mms21* gene-specific primer pairs. Here, a 17-bp insertion was found within exon 6 after the Pro180 codon that introduced six additional amino acid codons followed by a stop codon; if translated, the *mms21-2* protein would be missing most of the SP-RING domain and the C-terminal end (Fig 1C, 1D and 1F). While it remained possible that the 208-residue mms21-1 protein retained some of its activity, the 183-residue mms21-2 protein should be poorly functional given its compromised SP-RING domain. As judged by qRT-PCR analysis of homozygous plants, the *mms21-1* and *mms21-2* lines accumulated only 35% and 2%, respectively, of the *Mms21* mRNA level found in W22 (Fig 1E).

After backcrossing the *mms21-1* and *mms21-2* lines five times into the W22 inbred followed by self-pollination, homozygous plants were identified that displayed numerous growth defects as compared to their normal siblings. These defects segregated as recessive traits, with the phenotypic severity stronger for the *mms21-2* allele in agreement with its more compromised gene architecture and expression. While viable, homozygous *mms21-1* and *mms21-2* seeds germinated poorly and the seedlings grew more slowly, ultimately developing into severely stunted plants (as judged by fresh and dry weights; S1A Fig) with shorter roots and fewer leaves at maturity (Fig 2A–2D). *mms21-1/2* leaves were also significantly shorter and narrower but developed similarly-sized epidermal cells as compared to W22, suggesting that the growth defects arose from fewer cell numbers and not reduced cell expansion (S1B, S1C, and S1D Fig). To confirm that the phenotypes associated with the two mutant alleles are caused by the same gene, we performed genetic complementation tests using reciprocal crosses between the two *mms21* alleles. As shown in Fig 2E, F1 progeny harboring both mutant alleles, as determined by genomic PCR, displayed the identical stunted shoot phenotype as single homozygous *mms21-1* or *mms21-2* plants.

Further analysis of floral organs revealed that the *mms21* mutations also compromise reproduction. The mutant cobs grew poorly with fewer ovules. While tassels did appear, they were underdeveloped and the emerging anthers often failed to open, and even if opened, they shed much less pollen (Figs 2F, 2G and S2B). Silk emergence was also substantially delayed (S2A Fig), consistent with a delay in overall development. Germination assays on the small amount of *mms21-1/2* pollen collected revealed a substantial block in pollen tube emergence (Fig 2H and 2I), which when combined with other defects in cob and tassel development, likely underpinned the low fecundity seen for *mms21-1/2* plants. Finally, the resulting *mms21* seeds were slightly smaller in size, and often showed a pitted surface as compared to their normal siblings (Fig 2J and 2K). Approximately 25% of the seeds from self-pollinated *mms21-1* and *mms21-2* cobs acquired this appearance consistent with a recessive trait. Dissection of the mature seeds from self-crossed heterozygous *mms21-1/2* cobs revealed reductions in embryo size and often an underfilled endosperm, which were stronger for the *mms21-2* allele (Fig 2L–2N). This seed phenotype could be seen as early as 12-days-after pollination (DAP) for *mms21-2*, suggesting that MMS21 is important early in maize seed development (S3 Fig).

To further link the mutant phenotypes with the *Mms21* locus and potentially isolate stronger alleles, we generated additional germplasm by CRISPR/Cas9-mediated mutagenesis of the

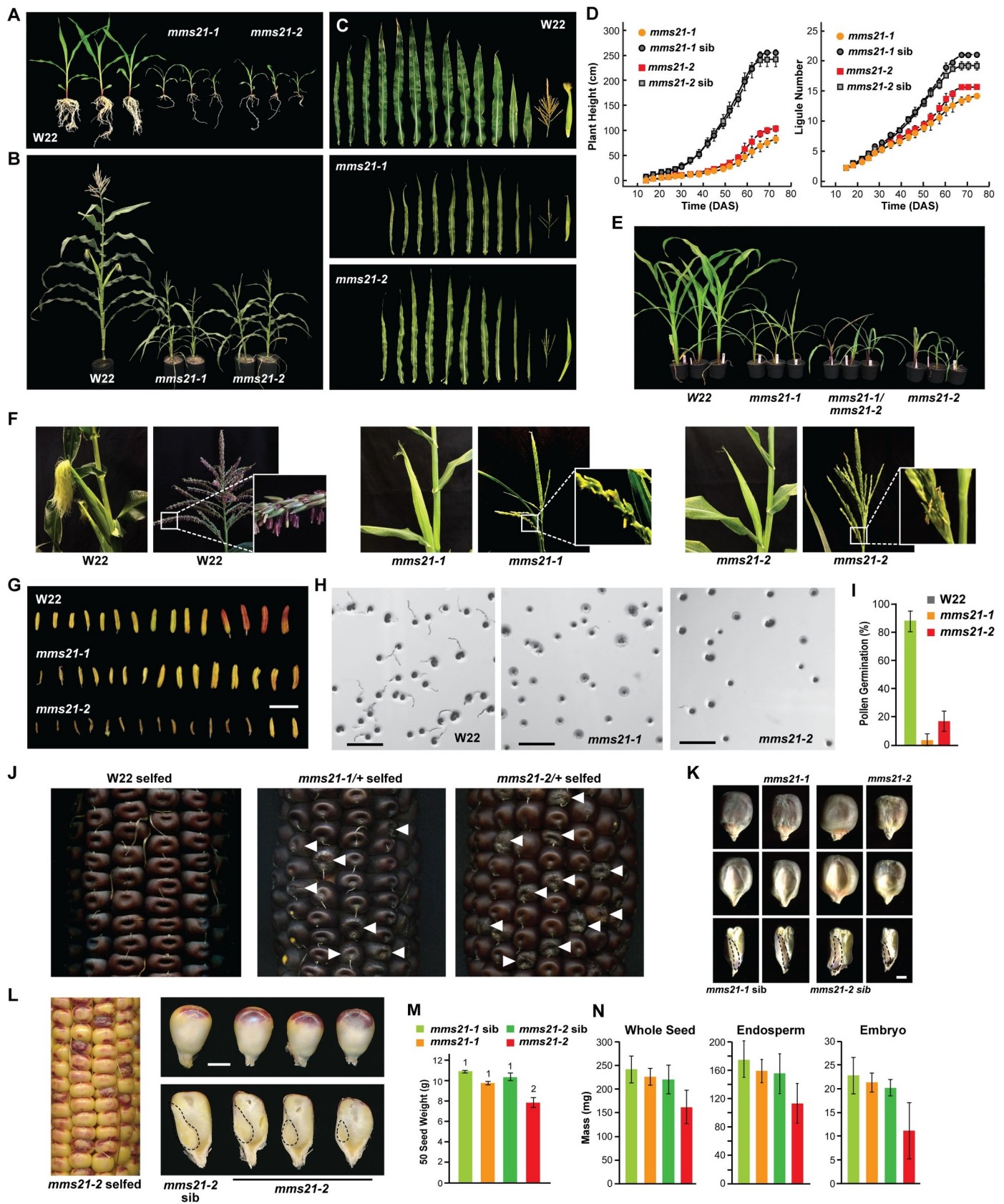

**Fig 2. Maize *mms21* Mutants Display Defects in Vegetative Growth and Reproduction. (A)** W22 and homozygous *mms21* plants imaged 14 DAS. The UniformMu *mms21-1* and *mms21-2* alleles are described in Fig 1. **(B)** W22 and homozygous *mms21* plants imaged at maturity (70 DAS). **(C)** Chronological display of the leaves, tassels, and ears from individual W22, *mms21-1*, and *mm21-2* plants at maturity. **(D)** Quantification of plant height and ligule number over time as measures of developmental progression. Each bar represents measurements of 4 plants (±SD). **(E)** Allelism test showing that transheterozygous progeny, generated by crossing the two mutant alleles, are phenotypically similar to plants homozygous for *mms21-1* or *mms21-2*. **(F)** Flowering is impaired in homozygous *mms21-1* and *mms21-2* plants. Shown are tassels and cobs at maturity. *mms21* mutants have reduced silking and closeup of anthers show limited anthesis from tassels. **(G)** Layout of representative anthers obtained from W22 and mutant tassels. *mms21-2* anthers are mostly aborted. Scale bar = 8 mm. **(H)** *mms21* pollen germinates poorly. Shown are pollen from W22, *mms21-1*, and *mm21-2* anthers incubated on germination medium for 3 hr at 28˚C. Scale bar = 1 mm. **(I)** Quantification of pollen germination efficiency shown in (H). Each bar represents the average of three biological replicates (±SD), each measuring at least 100 pollen grains. **(J)** Mature ears from self-pollinated W22 and *mms21/+* heterozygous plants showing the appearance of defective seeds (arrowheads). **(K)** Abnormal morphology of *mms21-1* and *mms21-2* seeds as compared to their normal siblings. The abgerminal (top), germinal sides (middle), and the saggital sections (bottom) of a representative seed are shown. Scale bar = 2 mm. **(L)** *mms21-2* seeds are smaller. Shown are whole seeds and a sagittal plane section showing variability in embryo size and possibly premature endosperm starch accumulation in seeds harvested from self-pollinated *mms21-2/+* plants at 16 DAP. Scale bar = 3 mm. **(M)** *mms21* seeds weigh less than their normal siblings. Each bar represents the weight of 50 seeds (±SE) from three biological replicates obtained from *mms21-1* and *mms21-2* and their heterozygous siblings collected at maturity. **(N)** Quantification of weight of seeds, endosperm, and embryos dissected from self-pollinated W22, *mms21-1*, and *mms21-2* cobs at 24 DAP. Each bar represents the average (±SD) of 50 seeds.

Hi-II background. The resulting seven *mms21-CR* mutations harbored an array of defects around the two Protospacer Adjacent Motif (PAM) sequences designed within the CRISPR target sites just upstream of introns 1 and 2 for Cas9 cleavage (Figs 1A,1C and 3A). Included were a 1-bp deletion before the 2nd PAM for *mms21-CR1*, a 14-bp deletion around the 2nd PAM for *mms21-CR2*, an additional A nucleotide before the 2nd PAM for *mms21-CR3* which likely shifted the reading frame to begin at an ATG codon located at the front of exon 3, an additional A nucleotide before the 1st PAM combined with 2-bp ΔCG deletion before the 2nd PAM for *mms21-CR4*, 6- and 8-bp deletions before the 1st and 2nd PAMs, respectively, for *mms21-CR5*, a 3-bp ΔAGT deletion before the 1st PAM for *mms21-CR6*, and a 33-bp deletion at the 1st PAM for *mms21-CR7* that removed 11 codons within exon 1 (Fig 1C and 1F). All these mutations altered the *Mms21* coding region by deleting one to as many as 200 residues if translated (Fig 1C).

Overall, our phenotypic analysis of the CRISPR/Cas9-derived collection after two back-crosses to the B73 inbred showed similar developmental consequences as seen for the Uni-formMu alleles, which included abnormal root, shoot and seed development, and pitted seeds

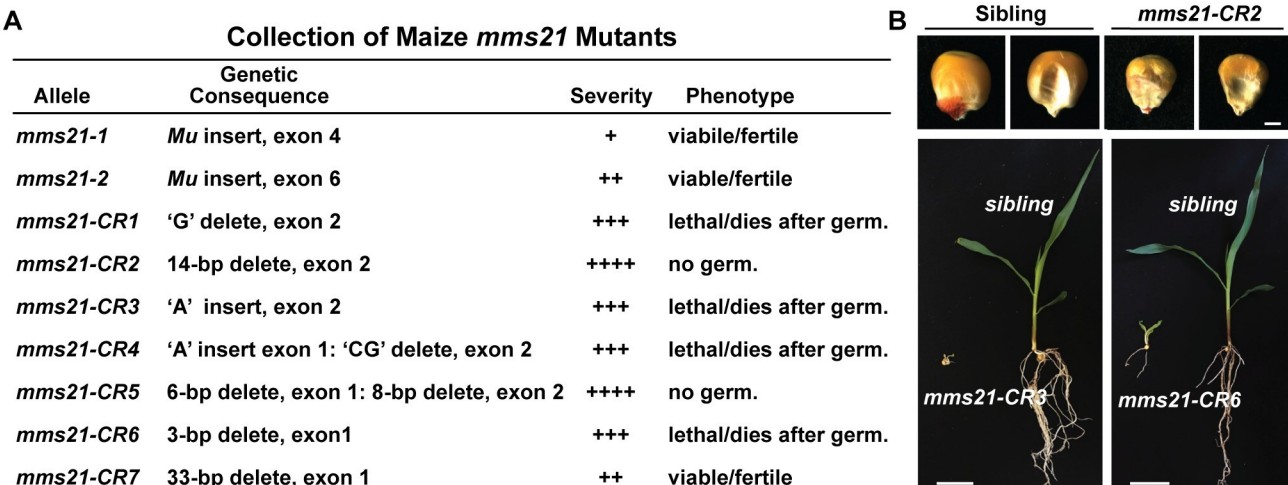

### A    Collection of Maize *mms21* Mutants

| Allele | Genetic Consequence | Severity | Phenotype |
|---|---|---|---|
| *mms21-1* | *Mu* insert, exon 4 | + | viable/fertile |
| *mms21-2* | *Mu* insert, exon 6 | ++ | viable/fertile |
| *mms21-CR1* | 'G' delete, exon 2 | +++ | lethal/dies after germ. |
| *mms21-CR2* | 14-bp delete, exon 2 | ++++ | no germ. |
| *mms21-CR3* | 'A' insert, exon 2 | +++ | lethal/dies after germ. |
| *mms21-CR4* | 'A' insert exon 1: 'CG' delete, exon 2 | +++ | lethal/dies after germ. |
| *mms21-CR5* | 6-bp delete, exon 1: 8-bp delete, exon 2 | ++++ | no germ. |
| *mms21-CR6* | 3-bp delete, exon1 | +++ | lethal/dies after germ. |
| *mms21-CR7* | 33-bp delete, exon 1 | ++ | viable/fertile |

**B** Sibling    *mms21-CR2*

**Fig 3. Strong *mms21* Alleles Generated by CRISPR/Cas9 Lead to Seedling Lethality. (A)** Comparisons of *mms21* mutant alleles derived from *Mutator* and CRISPR/Cas9-mediated mutagenesis. The positions of the mutations and their phenotypic consequences are indicated. **(B)** Representative CRISPR/Cas9-derived *mms21-CR* mutants showing the seed and seedling phenotypes as compared to their normal siblings. The seedlings were grown for 14 DAS.

with smaller embryos and dampened germination (Figs 3B and S4). Surprisingly, only the *mms21-CR7* allele was fertile when homozygous (S4C and S4D Fig) and resembled the *Mutator*-derived lines, while most of the remaining mutants failed to produce viable offspring, thus requiring maintenance as heterozygotes. The strongest phenotypic mutants were the *mms21-CR2* and *mms21-CR5* alleles; they produced homozygous seeds, but these seeds failed to germinate (S4A and S4B Fig). The next strongest impact was seen for the *mms21-CR1*, *mms21-CR3 and mms21-CR6* alleles that produced germinable seeds but the homozygous seedlings invariably died within approximately two weeks of growth under normal greenhouse conditions (Figs 3B and S4A–S4C). The *mms21-CR3* mutant was the strongest allele among the three and stalled growth soon after radical emergence (Fig 3B). qRT-PCR analysis of the six mutants that germinated (*i.e.*, all but *mms21-CR2* and *mms21-CR5*), using primers that spanned the coding region for part of the SP-RING domain (see S1 Table), showed that, with the exception of *mms21-CR7*, all strongly dampened accumulation of the *Mms21* mRNA as compared to their normal siblings (Fig 1E). The most intriguing homozygous-lethal allele was *mms21-CR6* as its 3-bp deletion effectively suppressed accumulation of the *Mms21* transcript even though its mRNA was in frame and missing only a single Val23 codon, suggesting that this region influences mRNA stability (Fig 1F). This severity for *mms21-CR6* was in contrast to the *mms21-CR7* allele, which was missing a larger portion of *Mms21* transcript but was still expressed at reasonable levels.

## *mms21* mutants have relatively normal SUMOylome profiles

As one strategy toward understanding how MMS21 affects maize, we assessed its overall impact on SUMOylation by subjecting total tissue extracts to immunoblot analysis with anti-SUMO antibodies [24]. As shown in Fig 4A, SUMO-conjugate profiles in *mms21-1* and *mms21-2* leaves were mostly indistinguishable from those seen in W22 both before and after a 30-min heat stress at 42˚C, which dramatically increases the pool of SUMO conjugates [24]. No species were absent in *mms21-1/2* leaves with or without the heat stress, and at most, only a few new species at ~60 and 37 kDa appeared. Similarly, we tested embryo and endosperm tissue harvested from seeds at 16 DAP (Fig 4B). Again, little differences in the profiles and levels of SUMO conjugates and free SUMO were detected in the mutants, strongly suggesting that MMS21 modifies only a small subset of SUMO substrates in maize, consistent with similar studies with Arabidopsis [5].

## *mms21* mutants strongly alter the maize transcriptome

Alternatively, we analyzed the maize transcriptome anticipating that the phenotypes of the *mms21* mutant collection were underpinned by robust, informative changes in gene expression. Here, we subjected total RNA isolated from *mms21-1* and *mms21-2* shoots harvested 10 days-after-sowing (DAS), and embryos and endosperm at 16 DAP to in-depth profiling by RNA-seq analysis, using equivalent samples from wild-type W22 for the comparisons. Each tissue was analyzed by three biological replicates (27 total samples) using paired-end reads of at least 150 bp, which ultimately resulted in ~47 M reads on average per sample. Approximately 80% of the high-quality reads could be uniquely mapped onto the maize B73 reference genome (RefGen_V4.48), which resulted in 28,657, 27,062 and 24,977 unique transcripts in the shoot, embryo and endosperm samples, respectively. For every genotype/tissue group, the replicates showed strong correlations in mRNA abundances (0.98 to 0.99) based on $\log_2$-transformed transcript read counts, thus confirming the reliability of the data (S5 Fig), which was also supported by principal component analyses (PCA) of the transcriptomes in which the mutant samples invariably clustered together and away from those of W22 (Fig 5A).

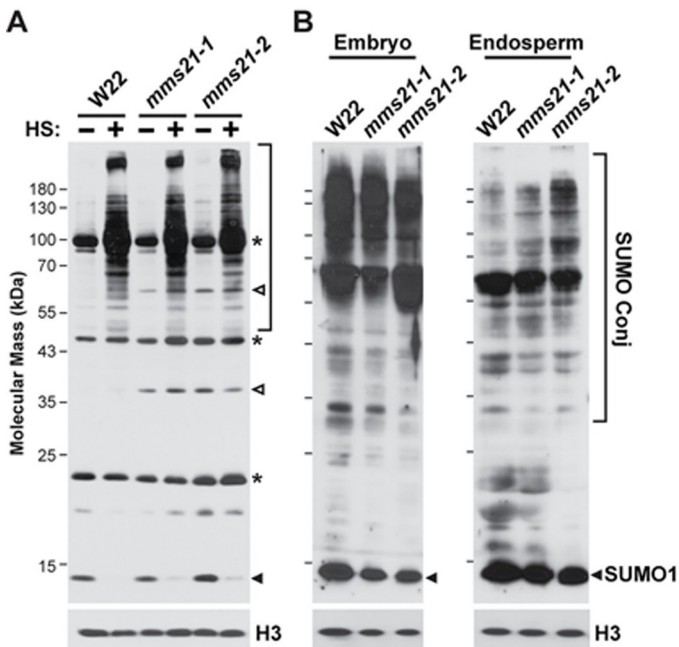

**Fig 4. Profiles of SUMO and SUMO Conjugates are Weakly Altered in *mms21* Mutants.** Total protein was extracted from the indicated tissues and subjected to SDS-PAGE and immunoblot analysis with anti-SUMO1 antibodies. Near equal protein loading was confirmed by immunoblotting with anti-histone H3 antibodies. Closed arrowheads and brackets locate free SUMO and SUMO conjugates, respectively. Open arrowheads locate proteins that differentially accumulated in the *mms21* backgrounds. Asterisks locate unidentified species that react with the anti-SUMO1 antibodies. **(A)** SUMO conjugates accumulating in seedling leaves before (-) or after (+) a 30-min heat shock at 42°C. **(B)** SUMO conjugate profiles in embryo and endosperm tissues collected at 16 DAP.

To further confirm the reliability of the datasets, we compared the expression profiles of significantly impacted genes (adjusted *p*-value <0.05) by fold change (FC) in abundances for each transcript. As shown in Fig 5B, strong correlations were seen between the *mms21-1* and *mms21-2* datasets versus W22 with correlation values for the $\log_2$-transformed data of 0.94, 0.83 and 0.87 for the shoot, embryo and endosperm samples, respectively, implying that the mutant transcriptomes responded similarly. Between W22 and the *mms21-1/2* mutants, 7,214 significant differentially expressed genes (DEGs; adjusted *p*-value < 0.05 based on multiple comparisons) were identified in shoots (3353 up-regulated transcripts and 3861 down-regulated), 5,864 in embryos (2,943 up-regulated transcripts and 2,921 down-regulated), and 1,320 in the endosperm (401 up-regulated transcripts and 919 down-regulated), which collectively indicated strong and pervasive changes in mRNA profiles for the mutants (Fig 5C). DEG comparisons among the biological replicates also revealed that the transcriptional responses of the *mms21-1* and *mms21-2* mutants were markedly similar. The impacted genes in common between the *mms21-1* and *mms21-2* samples represented a major fraction of the total DEGs in shoots but more minor fractions of the DEGs in embryos and endosperm (Fig 5D).

As one approach to identify the functional classes of genes whose expression were altered by the *mms21* mutations, we subjected the DEGs to Gene Ontology (GO) analysis using the Annotation Hub database [35]. Enrichment analysis detected numerous GO categories that spanned many aspects of plant growth and development, including 'sexual reproduction', 'cellular catabolic', 'proteasomes', 'nucleosomes', 'chromatin', 'vacuoles', 'ER', and 'peptidases' for the upregulated genes, and numerous photosynthesis-related categories for the downregulated genes (S6A–S6C Fig). This conclusion was corroborated by Kyoto Encyclopedia of Genes and

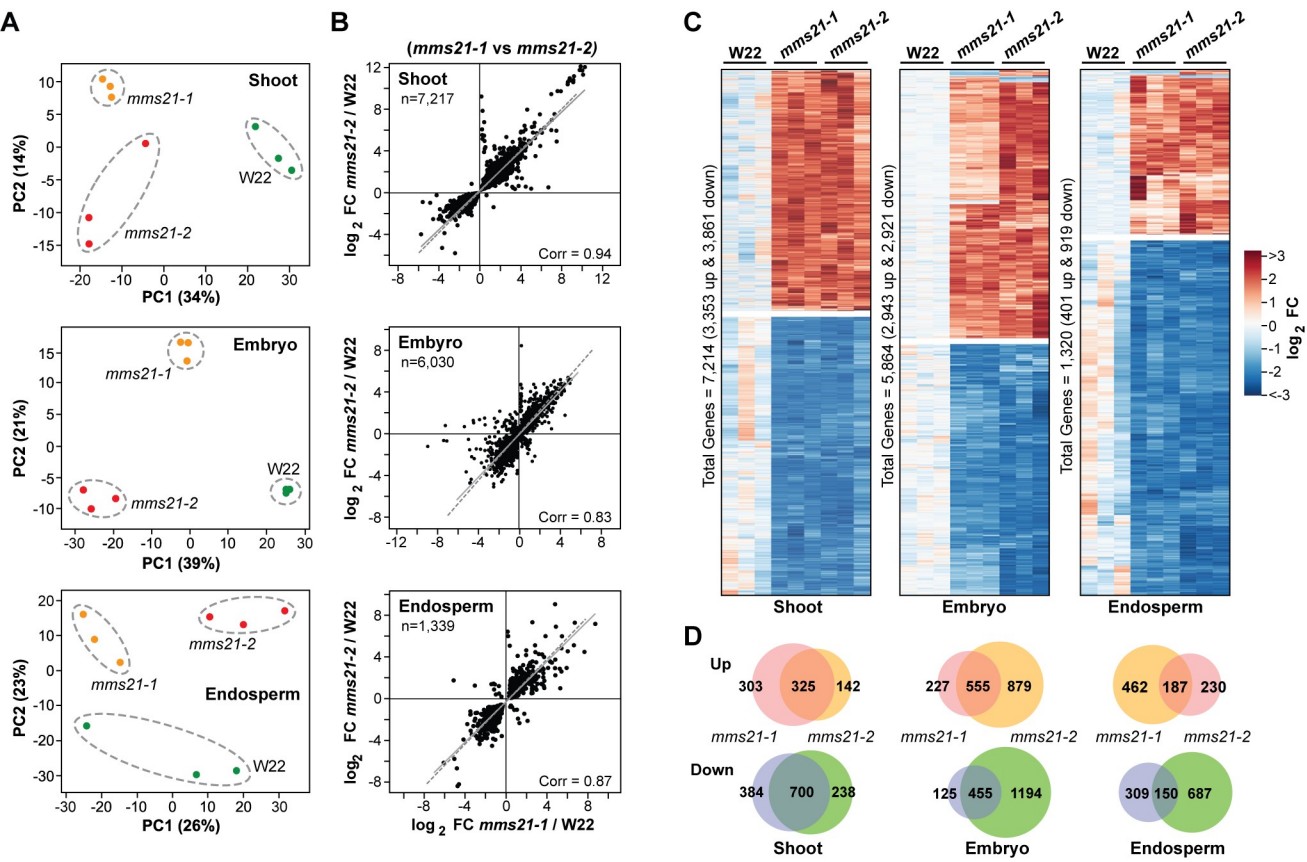

**Fig 5. *The mms21-1* and *mms21-2* Mutations Profoundly Alter the Maize Transcriptome.** mRNA profiles were generated by RNA-seq from W22, *mms21-1*, and *mms21-2* shoots collected at 10 DAS, and from embryos and endosperm isolated at 16 DAP. **(A)** PCA of the transcriptome data sets showing that the *mms21-1* and *mms21-2* mRNA profiles differ from W22. The values were determined from log$_2$-transformed transcript counts for each dataset (n = 3 biological replicates). The dashed lines outline biological replicates associated with each genotype. **(B)** Scatterplots of significantly affected transcripts (adjusted *p*-value <0.05) in the two *mms21* alleles versus those in W22. The total numbers of significantly affected transcripts analyzed are indicated, along with Pearson's correlation coefficient (Corr) values. Solid lines show the correlation within each comparison; dashed lines show correlations equal to 1. Each point represents the mean of three biological replicates. **(C)** Heat maps showing the read counts for shoot, endosperm, and embryo transcripts significantly affected by each *mms21* allele, as determined by DESeq2 using an adjusted *p*-value <0.05. The abundances for each allele were normalized to the average values obtained from W22. **(D)** Venn diagram showing the overlap of transcripts significantly affected in *mms21* versus W22 shoots, embryo and endosperm as determined by the adjusted *p*-value < 0.05 and a FC ≥2-fold up or down.

Genomes (KEGG) analyses [36], where 'sugar metabolism', 'protein processing in ER', 'photosynthesis', and 'photosynthesis-antenna proteins' emerged as enriched categories for the *mms21* mutants (S6D Fig). Overall, our RNA-seq data implied that the MMS21 globally influences much of the maize transcriptome.

To hone in on the transcriptional response of specific genes, we analyzed the RNA-seq datasets by volcano plots that compared the FC and *p*-value of significance for the DEGs between W22 and the *mms21-1/2* mutants (Figs 6A and S7). The exaggerated splay of the plots revealed that the abundances of many transcripts were significantly altered (24.9%, 22.4% and 5.4% of total transcripts in shoots, embryos and endosperm, respectively), some by as much as a thousand FC (log$_2$ FC >10). When we focused on the SUMOylation pathway, expression of almost all identified components were unaffected by the mutations with the exception of the *Mms21* mRNA itself whose abundance was strongly reduced (Figs 6A and S7). These observations not only confirmed that the mutant alleles dampened *Mms21* expression, but also implied that other aspects of SUMOylation were not upregulated as compensation.

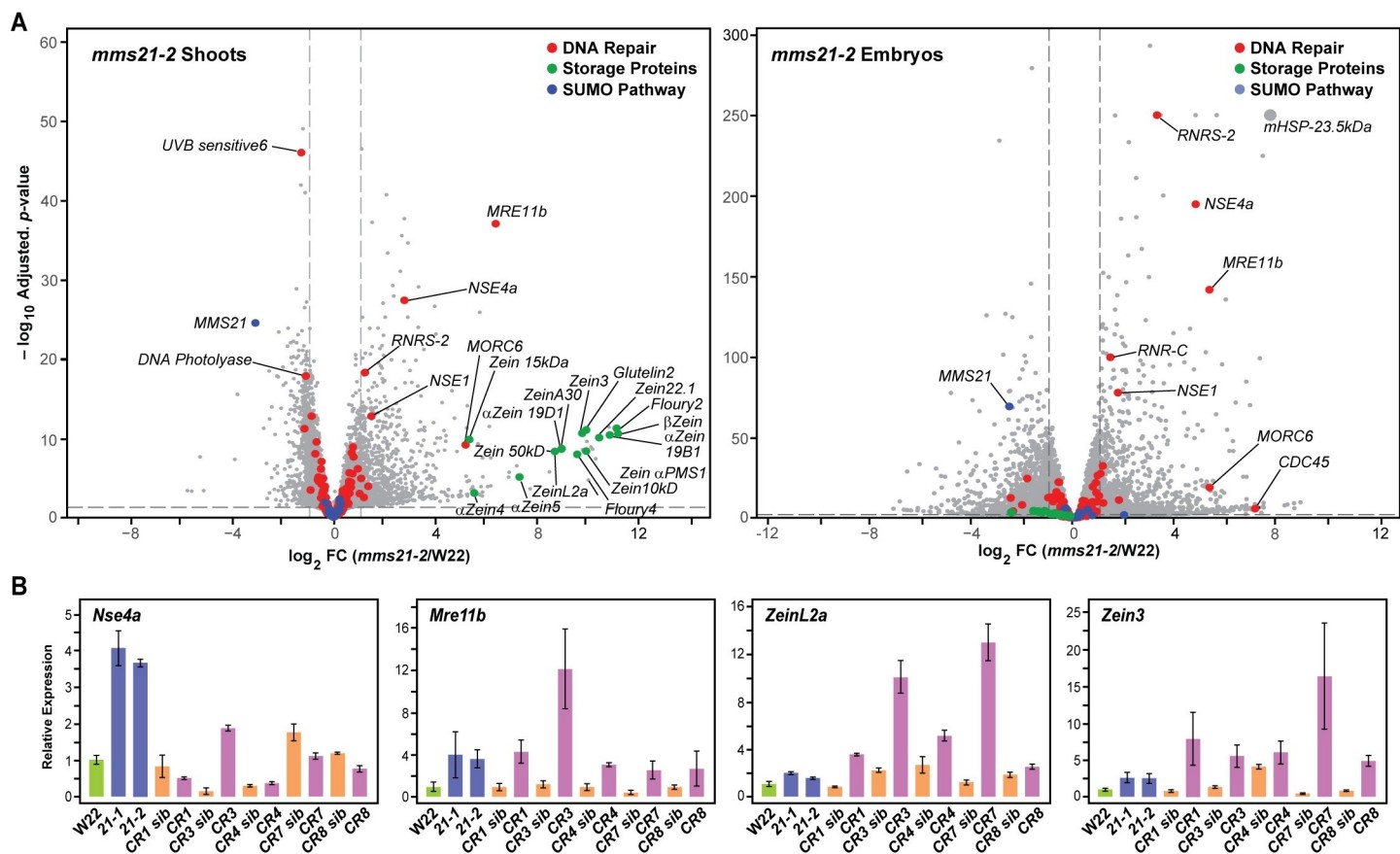

**Fig 6. MMS21 Strongly Influences the Expression of a Subset of the Maize Transcriptome. (A)** Volcano plot representation of DEGs in the *mms21-2* shoots and embryos as compared to those in W22 based on log$_2$ FC and -log$_{10}$ adjusted *p*-values. Blue, red, and green dots identify genes involved in the SUMO pathway and DNA repair, or encode seed storage proteins, respectively. Specific DEGs of interest are noted. **(B)** Confirmation of select DEGs shown in panel (A) by qRT-PCR analysis of the collection of *mms21* mutants generated by UniformMu or CRISPR/Cas9-mediated mutagenesis (see Fig 1). The results from the CRISPR/Cas9 mutants were compared to their normal siblings. Each bar represents the mean of three biological replicates (±SD). See S7 Fig for similar volcano plots analyzing the transcriptomes of *mms21-1* shoots, *mms21-1* embryos and endosperms for both alleles.

Given the likely connections between MMS21 and DNA repair, we also interrogated the volcano plots for genes associated with this process (Fig 6A). Notably, most DNA repair-associated mRNAs were modestly impacted (either up or down) using a FC cutoff ≥2 and an adjusted *p*-value <0.05. Intriguingly, the exceptions were mRNAs for *Nse4a* and *Nse1* (annotated as embryo-defective 1379 in reference genome RefGen_V4.48) that encode components of the SMC5/6 complex involved in DNA repair and chromatin stability (Figs 6A and S7). While their mRNA abundance was robustly elevated by the *mms21* mutations, those encoding the central SMC5 and SMC6 subunits and four other maize *Nse4* paralogs were not altered, suggesting the upregulation of *Nse1* and *Nse4a* is unique among *Smc5/6* complex genes.

Additional upregulated loci in the *mms21* backgrounds connected to genome integrity were *Mre11b*, which encodes a core component of the Mre11/Rad50/Nbs1 complex that functions in DNA repair and is critical for maintaining genome stability in Arabidopsis [37], and *MICRORCHIDIA* (*Morc*)-6, which encodes a relative of the Arabidopsis MORC transcriptional repressor family that associates with nuclear bodies and globally mediates transcriptional silencing [38,39] (Figs 6A and S7). mRNAs encoding the small subunits of ribonucleoside diphosphate reductase (RNR), RNRS-2 and RNRS-C, that participate in DNA repair via the synthesis of deoxyribonucleotides [40] were also significantly enriched in the

*mms21* datasets. In addition, genes encoding several protein chaperones (HEAT SHOCK PROTEIN (HSP)-70, HSP-23.5kDa, and the class-II small HSP17.5-kDa) were upregulated in some of the *mms21-1/2* samples that could reflect enhanced proteotoxic stress.

The most interesting DEGs encoded zein storage proteins, which displayed a pronounced expression in *mms21-1/2* shoots, a tissue that normally does not express these proteins. While this effect was not evident in embryos and endosperm, the latter of which naturally accumulates high levels of zeins during development [41], the response was substantial in leaves and, in fact, represented the most significantly impacted collection of mRNAs in the *mms21-1/2* datasets (Figs 6A and S7). Of the 54 transcripts showing a ≥32 FC in abundance for *mms21-2* versus W22 shoots, 37 encoded zeins. Interestingly, when we assayed two general classes of zein proteins (α- and γ-zein) by immunoblot analysis with class-specific antibodies [41], we failed to detect such proteins in leaves despite their ectopic mRNA accumulation (S8 Fig).

To confirm these changes in mRNA levels *for Nse4a*, *Mre11b*, and two representative zein genes (*ZeinL2a* and *Zein3*), we quantified their transcript abundances by qRT-PCR using our complete collection of *mms21* mutants generated by UniformMu and CRISPR/Cas9-mediated mutagenesis. As shown in Fig 6B, the majority of these genes were uniformly upregulated in the *mms21* mutants but often not to the same robust levels as those seen by RNA-seq. The only outlier was *Nse4a* in some of the CRISPR/Cas9 mutant alleles; the abundance of this mRNA varied widely as compared to the values obtained for its siblings, which might reflect variations in allele severity and the hybrid state of the CRISPR/Cas9 backgrounds.

## *mms21* mutations alter the transcriptome/proteome balance

To further assay the impact of MMS21 on protein accumulation, we compared the proteomes of *mms21* mutant and normal siblings by shot-gun mass spectrometry (MS) of young seedlings [42]. Here, trypsinized total protein extracts from 10-DAS seedlings were subjected to reversed-phase separation followed by tandem MS, which allowed relative quantification for approximately 4,000 proteins from the MS1 scans based on the mean of three biological replicates each with two technical replicates. The proteome data were then normalized among samples using a list of 150 proteins relatively unaffected by the mutations [42], which was then validated by assaying the levels of abundantly detected histones which should be consistent across genotypes (Figs 7A, S9A and S9B).

When the normalized proteome data were displayed by volcano plots that assessed both $\log_2$ FC in abundance versus *p*-values of significance, numerous proteins were significantly more or less abundant in the *mms21* seedlings compared to their normal siblings based on a FC = 2 threshold and adjusted *p*-values <0.05. For the *mms21-2* background, the values were 476 up- and 569 downregulated (26% mis-accumulated), while for the *mms21-1* background, the values were 496 up and 640 downregulated (27% mis-accumulated) as compared to 9% of the proteins having different values (>2 FC, *p*-value <0.05) when comparing normal siblings of *mms21-2 and mms21-1* to each other (Figs 7A, S9A and S9B). (We presume that most of the proteins assigned as differentially accumulating between the normal siblings represent noise inherent to MS data collection and analysis.) As with the immunoblot assays, we failed to detect any zein storage proteins by MS in the *mms21* seedlings despite having elevated mRNAs. When the differentially accumulating list was subjected to GO analysis, a broad spectrum of protein functionalities was impacted in the *mms21* backgrounds, such as 'cellular process', 'metabolic process' and 'catalytic activity', with little selective impact seen on specific subcategories, thus likely reflecting a general alteration in protein composition (Fig 7B).

We then compared the differences in protein abundance for the *mms21-2* seedlings versus normal siblings with those described above for the corresponding transcripts for 3,756

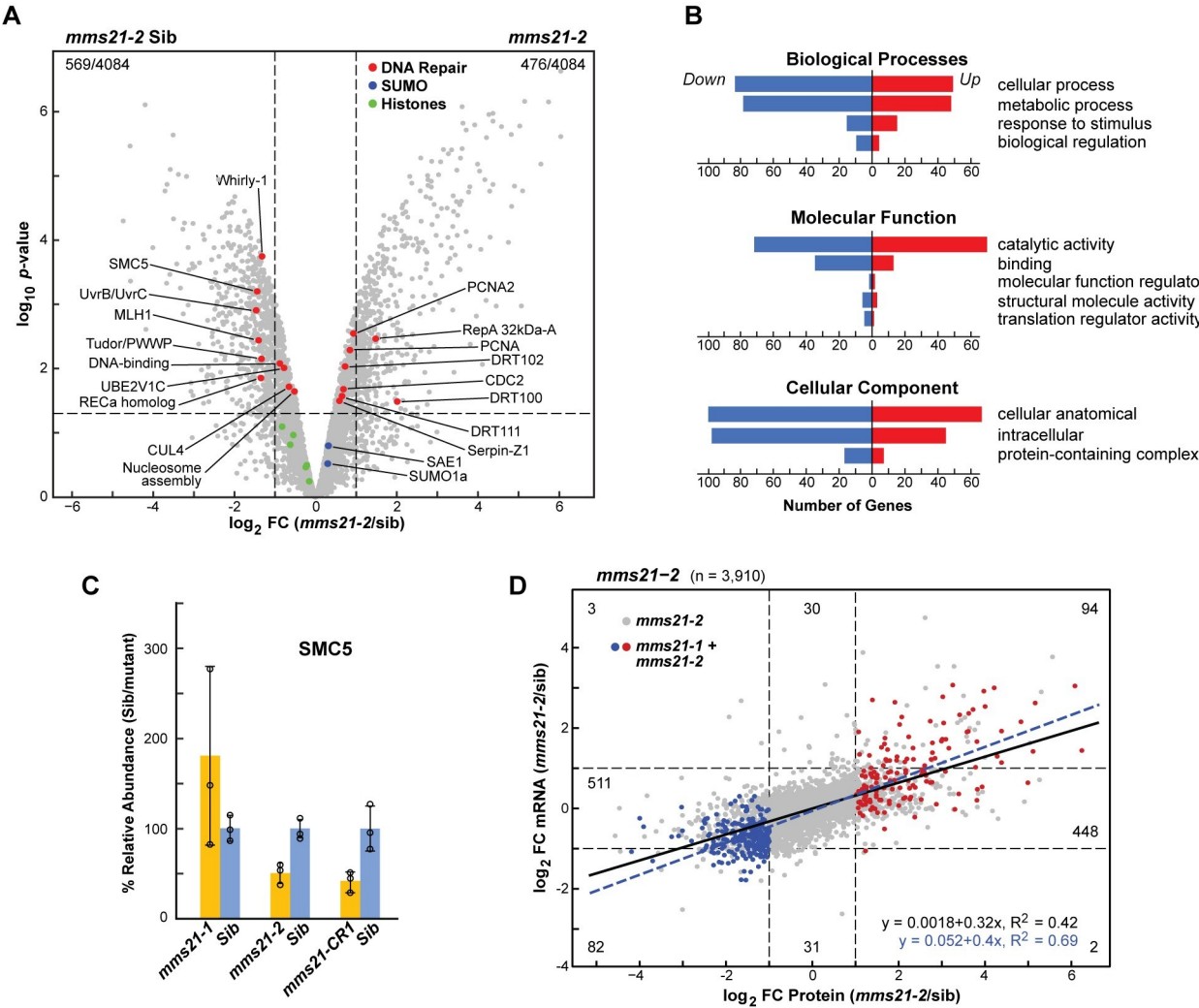

**Fig 7. *mms21* Mutants have Altered Proteome Profiles and Reduced SMC5 Levels. (A)** Altered proteome profile for the *mms21-2* mutant. The volcano plot depicts protein abundance changes for 4,084 proteins detected from *mms21-2* leaves as compared to those of its normal sibling. Each dot represents one protein that had detectable expression in both samples and was plotted based on its $\log_2$ FC in abundance (mutant/normal siblings) and its $-\log_{10}$ $p$-value of significance based on the three biological replicates, each with two technical replicates. The horizontal and vertical dashed lines mark a FC = 2 in protein abundance and a $p$-value = 0.05, respectively. Histone proteins used to confirm data normalization are shown as green. SUMO pathway components and DNA repair-associated proteins are highlight in blue and red respectively. **(B)** GO analysis of significantly regulated proteins in the *mms21* mutant versus W22 seedlings. The vertical coordinates indicate the enriched GO terms, and the horizontal coordinates show the number of genes for each GO term when comparing differentially expressed genes common between *mms21-1* and *mms21-2* seedlings. Negative values indicate downregulation, positive values indicate upregulation. GO enrichment was performed using all the three sub-ontologies: 'biological process', 'molecular function', and 'cellular component'. **(C)** SMC5 protein abundance is reduced in strong *mms21* mutant backgrounds. Relative protein abundances of SMC5 from the *mms21-1*, *mms21-2* and *mms21-CR1* mutants and their normal siblings. Each bar represents the mean of three biological replicates(±SD). **(D)** Positive correlation between transcriptome and proteome data in *mms21* mutants. Scatter plot showing the relationship between changes in protein and mRNA abundances for the *mms21-2* mutant versus normal sibling as determined by plotting the $\log_2$ FC in mRNA abundance versus the $\log_2$ FC in protein abundance. Red dots (154) and blue dots (277) highlight proteins that were more or less abundant (FC>2), respectively, in both *mms21-1 and mms21-2* leaves as compared to their normal siblings. The black and dashed blue lines show the correlations for *mms21-2* and for the combined data of *mms21-1* and *mms21-2* (red and blue dots).

proteins with data available for both. Surprisingly, a modest but significant correlation was seen ($R^2$ = 0.42), with the proteins less abundant in the mutant also having less mRNA, while those proteins more abundant in the mutant also having more mRNA (Fig 7D). This correlation was even more robust (0.69) when we analyzed only those proteins (431 total) that were

up or down-regulated in both the *mms21-1* and *mms21-2* backgrounds (Fig 7D). Collectively, the protein/mRNA correlations implied that the lack of MMS21 globally influenced the proteome balance primarily by altering the transcriptome balance.

## Role of MMS21 in DNA repair

To more specifically address a possible connection between MMS21 and DNA repair [13,43], we compared mRNA abundances for a number of likely contributors as identified by sequence homology to known Arabidopsis factors. From the analysis of this collection by Z-scores, it became apparent that the *mms21-1/2* lines had globally altered expression of DNA repair-associated genes, suggesting a dysregulation of the process. For example, of the 66 mRNA analyzed from embryos, many were upregulated in the mutant backgrounds, including those encoding a number of well-described DNA-damage repair factors such as PCNA, NSE1, NSE4a, MRE11b, BRCA1, and multiple subunits of the REPLICATION PROTEIN A (RPA) and RNR complexes, while others were downregulated, including those encoding the DNA mismatch repair protein MSH4 and DMC1 involved in meiotic recombination (Fig 8A and S2 Table). Similar responses were also seen for the few corresponding DNA repair proteins that we could detect by MS, but their measured changes were more muted with most failing to rise above/ below a FC = 2 cutoff (*p*-value<0.05). Only a few showed increases or decreases >2 fold in *mms21-2* seedlings, including the DNA mismatch repair protein MLH1, the UVRB/UVC homolog, DRT family members, a Tudor/PWWP ortholog, Whirly1, and SMC5 (Fig 7A).

Observing a potential connection of MMS21 to DNA repair, we next checked the sensitivity of the *mms21* mutants to DNA-damaging agents as assayed by the growth of emerging roots. Both *mms21-1* and *mms21-2* roots were strongly hypersensitive to MMS and mitomycin C, relatively unaffected by hydroxyurea, and slightly affected by bleomycin and zebularine but only for the stronger *mms21-2* allele (Fig 8B). The hypersensitivity of *mms21* to MMS in particular was consistent with the first discovery of this locus via a yeast MMS-sensitivity screen [44]. And finally, we measured the frequency of DNA strand breaks by comet assays [45]. Here, W22 and *mms21* nuclei were isolated from seedling roots, embedded in agarose, and then their DNA was electrophoresed under alkaline conditions; increased DNA breaks were then observed by a greater comet tail length as the DNA migrated toward the anode. As can be seen in Fig 8C and 8D, DNA from the *mms21* mutants had more breaks as compared to W22. Taken together with the RNA-seq data, we found that loss of MMS21 impacted DNA integrity and repair, along with substantially altered gene expression that included the ectopic expression of zein-encoding loci.

Prior studies with Arabidopsis *mms21/hpy2/nse2* mutants implicated MMS21 in endoreduplication, with the mutant shoot nuclei often harboring excess whole genome duplications [12] and pollen having a high frequency of diploid male gametes [29]. When similarly assessed for DNA content by flow cytometry of nuclei, we detected a comparable distribution of 2N and 4N nuclei for 10-DAS *mms21-1* leaves as compared to those from wild-type W22, suggesting little impact of MMS21 on cell division in maize somatic tissues (S10A and S10B Fig). Given that endoreduplication is a common feature of maize seeds, especially in the endosperm that undergoes multiple rounds of DNA replication before maturation [46], we then examined the ploidy levels of seed nuclei, using the *mms21-2* allele which allowed us to visually discriminate homozygous mutant kernels from their wild-type siblings on the same cob (see S3 Fig). As expected, we detected an expanded series of ploidy levels in whole seeds at 16-DAP, which included 2N, 4N and 8N nuclei in embryos, with the 3N, 6N and 12N nuclei likely representing triploid endosperm nuclei (S10C and S10D Fig). Again, a comparable ploidy distribution was seen for the *mms21-2* seed tissues versus W22. Taken together, we were left to conclude

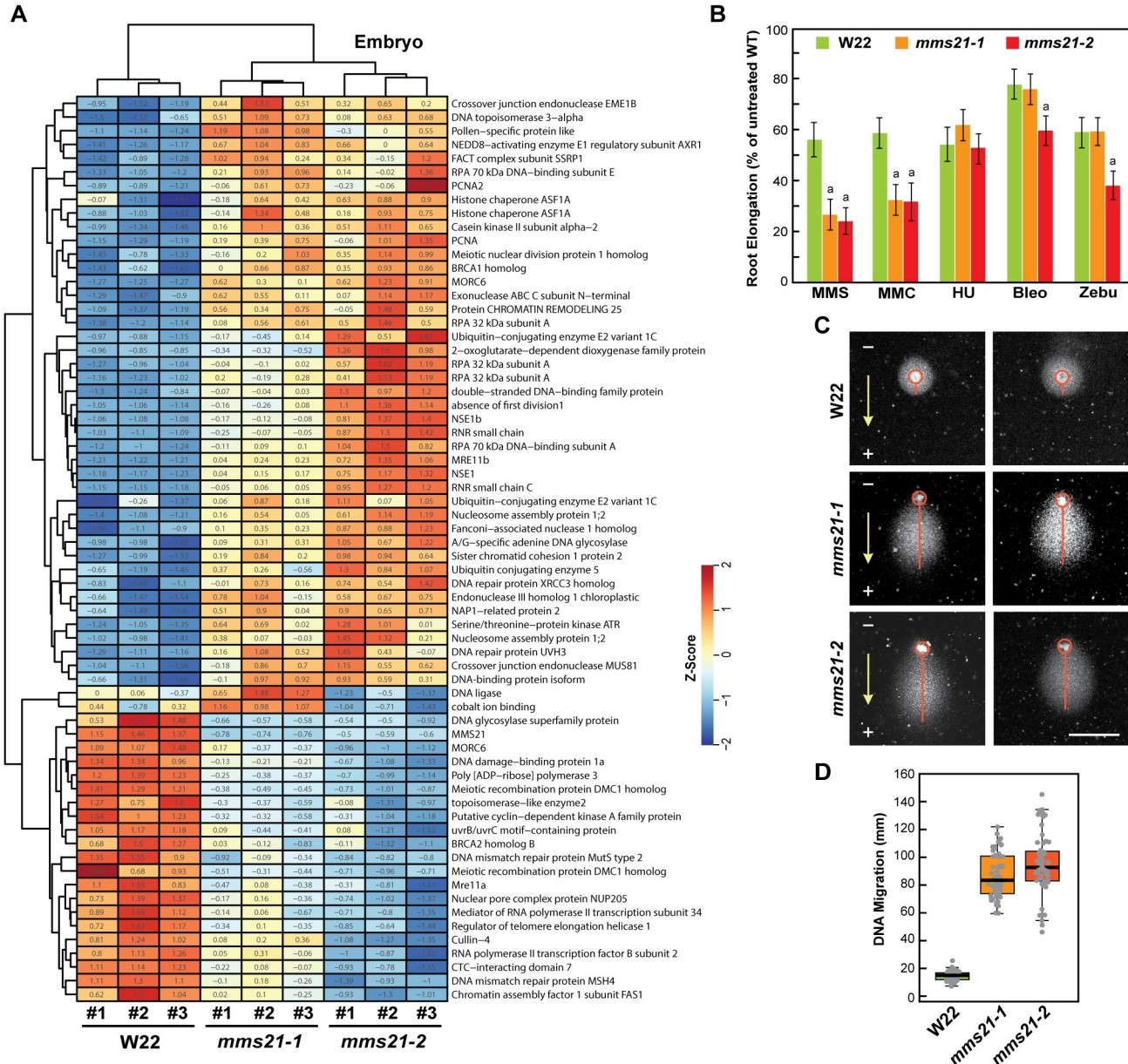

**Fig 8. *mms21* Mutants Have an Altered DNA Damage Response. (A)** Genes associated with DNA repair have altered expression in the *mms21* mutants. Shown are RNA-seq heat maps generated by Z scores from *mms21-1* and *mms21-2* embryos as compared to those from W22 focused on the altered expression of 66 genes associated with DNA damage repair. Each column represents an individual biological replicate; rows represent specific genes of interest. The numerical Z scores in each box shown standard deviations away from the mean. Groups of genes with similar expression patterns were clustered by columns and rows. **(B)** *mms21* seedlings are hypersensitive to DNA-damaging agents. *mms21-1*, *mms21-2*, and W22 seeds were germinated on sterile filter papers and then transplanted to solid growth medium supplemented with 20 ppm methyl methanesulfonate (MMS), 2 μM mitomycin-C (MMC), 1 mM hydroxyurea (HU), 100 nM bleocin (Bleo), or 10 μM zebularine (Zebu). After one-week, root growth, was measured and plotted relative to that seen with untreated W22 roots. Each bar represents the mean of three biological replicates (±SD). **(C)** Comet assays measuring by electrophoretic mobility the extent of DNA breaks. DNA from nuclei isolated from *mms21-1*, *mms21-2*, and W22 roots were subjected to agarose gel electrophoresis and then stained with propidium iodide. Mobility of the DNA was measured by the distance from the center of the nucleus to the edge of the comet tail. Scale bar = 50 μm. **(D)** Quantification of the comet assays in panel (C) by box plots based on the distribution of comet tail lengths for individual nuclei. The bottom and top of each box indicate the first (Q1) and third (Q3) quartiles, and the middle line reflects the median; the upper-limit equals Q3 plus 1.5 times interquartile range (IQR), and the lower-limit equals Q1 minus 1.5 times IQR. Each dot represents a single measurement (n = 50 cells).

that MMS21 has little impact in maize endoreduplication in contrast to that reported in Arabidopsis [12]. However, it is not yet known whether the ploidy levels of male gametes are affected.

## MMS21 interacts with SUMO, the SCE1 E2, and the SMC5 and NSE4a Subunits of the SMC5/6 complex

We presumed that MMS21 influences DNA dynamics, and ultimately maize development, through SUMOylation of one or more targets. Given: (i) the known connections between MMS21 and the SMC5/6 complex in Arabidopsis and other organisms [13,43], (ii) proteomic indications that Arabidopsis NSE4a is a MMS21 substrate [5], and our discoveries here in maize that (iii) *mms21* mutants are hypersensitive to DNA damage, and that (iv) the mRNAs encoding the NSE1 and NSE4a subunits of the SMC5/6 complex are selectively upregulated in *mms21* backgrounds, led us to speculate that MMS21 SUMOylates one or more components of the SMC5/6 complex. A further connection was evident when directly quantifying SMC5 protein levels in *mms21* seedlings by MS; as shown in Fig 7C, SMC5 levels were significantly lower in the strong *mms21* mutants, *mms21-2* and *mms21-CR1*, as compared to their normal siblings.

To provide further connections, we examined by yeast two-hybrid (Y2H) assays, whether MMS21 binds maize NSE4a and SMC5 using maize orthologs of the reported interactors BRAHMA and DPa as controls [30,31]. As shown in Fig 9A, MMS21 bound to one of its cognate E2s SCE1b (but not SCE1f) and to DPa, but only poorly to an N-terminal soluble fragment of BRAHMA as judged by growth on selection medium. None of the interactors bound SUMO1a. This association between MMS21 and SCE1b was lost when we used the expected polypeptides derived from the *mms21-1* and *mms21-2* alleles, implying that these aberrant forms poorly bind the SUMO-E2 intermediate. Intriguingly, MMS21 also bound strongly to SMC5 and NSE4a, with this association only weakly dampened when using the mms21-1 and mms21-2 protein variants (Fig 9A). Because MMS21 and NSE4a likely do not touch each other directly based on a general model of the SMC5/6 complex ([47]; see S14D Fig), we hypothesize that SMC5 helps tether these two proteins.

To further validate the MMS21 interactions *in planta*, we applied bimolecular fluorescence complementation (BiFC) assays that transiently expressed maize SUMO1a, SCE1b, SCE1f, SMC5 and NSE4a as fusions with the N-terminal and C-terminal fragments of YFP in *Nicotiana benthamiana* epidermal cells: interactions were then scored by reconstituted YFP fluorescence. Even though we failed to detect interactions between MMS21 and SUMO1a by Y2H, strong BiFC signals was evident in the nucleus and cytoplasm of leaf cells co-expressing MMS21 and SUMO1a (Figs 9B and S11), confirming the expectation that MMS21 interacts with SUMO1a *in planta*. Likewise, we detected BiFC interactions between MMS21 and SCE1b and now weak interaction of MMS21 with SCE1f in both the nucleus and cytoplasm. Most interestingly, *N. benthamiana* cells co-expressing MMS21 with NSE4a or SMC5 also reconstituted YFP fluorescence but these signals were only evident in the nucleus, consistent with the known nuclear location of the SMC5/6 complex (Fig 9B). While the mms21-1 protein appeared to retain its affinity for NSE4a and SMC5 based on the BiFC signals, this affinity appeared less strong for the mms21-2 protein, potentially in agreement with its more compromised architecture.

## SUMOylation of SMC5 by MMS21 *in vitro*

Our next objective was to demonstrate that MMS21 is a SUMO ligase that modifies SMC5 and/or NSE4. Here, we developed an *in vitro* SUMOylation system modified from Augustine

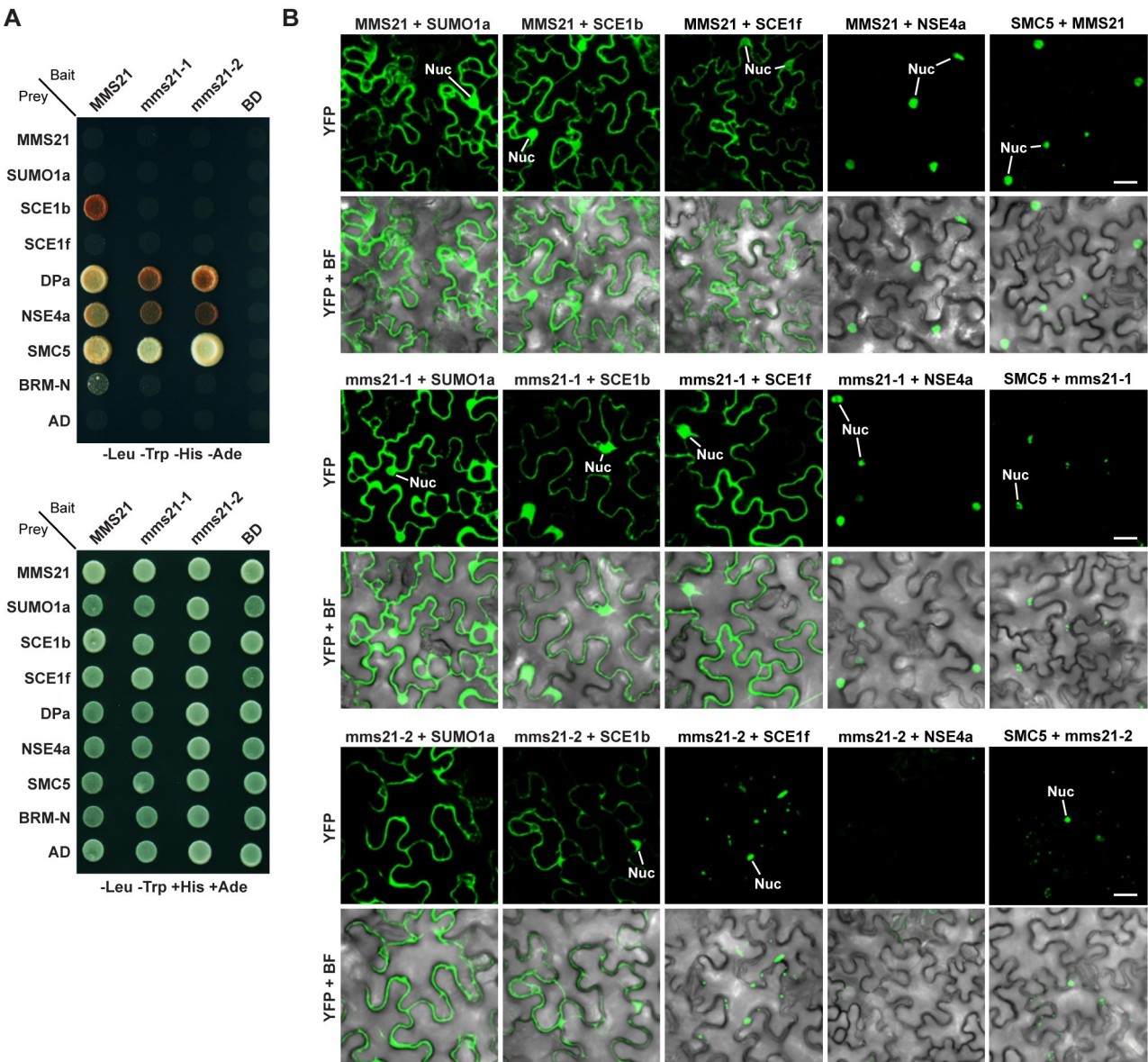

**Fig 9. Maize MMS21 Interacts with the SUMO Conjugation Machinery and Components of the SMC5/6 Complex. (A)** Y2H assays testing the interactions between full-length MMS21, or the mms21-1 and mm21-2 truncations, and various components within the SUMO pathway and the NSE4a and SMC5 subunits of the SMC5/6 complex. The known MMS21 interactors, DPa and an N-terminal soluble region of BRAHMA, were included for comparisons. All proteins were derived from maize and were expressed as N-terminal fusions with either the GAL4-activating domain (AD) or DNA-binding domains (BD). BD and AD represent empty vector controls. Shown are colonies grown for 3 d at 28˚C on selective medium lacking Leu, Trp, His and adenine (Ade) (top), or on non-selective medium missing only Leu and Trp (bottom). **(B)** BiFC assays testing pairwise the interactions between several partners shown in panel (A) and wild-type and mutant forms of MMS21. *N. benthamiana* leaf epidermal cells were co-infiltrated with plasmids expressing the N-and C-terminal fragments of YFP (nYFP and cYFP, respectively) fused to the indicated proteins. Reconstituted YFP fluorescence of epidermal cells along with bright field (BF) views were imaged 40–45 hr after infiltration. Tested pairs were nYFP-MMS21 with cYFP-SUMO1a, nYFP-MMS21 with cYFP-SCE1b or cYFP-SCE1f, nYFP-MMS21 with cYFP-NSE4a, and nYFP-SMC5 with cYFP-MMS21. Additional BiFC control images are found in S11 Fig. Note that MMS21 interacts in both the cytoplasm and nucleus with the SUMOylation machinery but only in the nucleus with NSE4a and SMC5. Nuc, nucleus. Scale bars = 20 μm.

*et al.* (2016), using recombinant maize proteins affinity purified via appended 6His tags. Ultimately, the system was built with SUMO1a, the SAE1/SAE2 E1 heterodimer, the SCE1b E2, and full-length MMS21 (S12 Fig). Specificity of the reactions for processed SUMO1a was

confirmed by using the unprocessed SUMO1a precursor bearing its C-terminal extension that blocks conjugation (UP-SUMO1a), processed and active SUMO1a with its exposed glycine needed for the isopeptide bond, and a K0 variant in which all 7 SUMO lysines were substituted for arginines and thus unable to assemble SUMO-SUMO chains [24]. In reactions with just SUMO1a, E1 and E2, the kinetics of SUMOylation was then optimized for time, and ATP and E2 concentrations, using the conversion of free SUMO1a into higher molecular mass adducts for the output as assessed by immunoblot assays with anti-SUMO1 antibodies (S13A and S13B Fig). When we added MMS21 to the E1 and E2-containing reactions, robust SUMOylation was clearly evident, which reached saturation within 1 hr as opposed to overnight reactions missing MMS21 (S13A and S13C–S13D Fig). To confirm the correct enzymatic scheme, we tested the impact of pyrophosphate, which interferes with formation of the SUMO-adenylate intermediate [48]. In reactions containing 2 mM ATP, SUMOylation was strongly suppressed by pyrophosphate concentrations at 0.1 mM or above (S13E and S13F Fig).

Using this optimized SUMOylation system, we confirmed that MMS21 is a *bona fide* E3 that requires ATP, processed SUMO1a, and E1 and E2 activities to generate SUMO adducts (Fig 10A). While MMS21 did not use unprocessed SUMO1a, conjugation was successful with the K0 variant although not to the same extent as the processed form, strongly suggesting that some conjugation reflected assembly of SUMO-SUMO chains. When the predicted mms21-1 and mms21-CR3 polypeptides harboring the SP-RING were used, conjugation was retained as expected if this domain is required for E2-SUMO binding (Fig 10B).

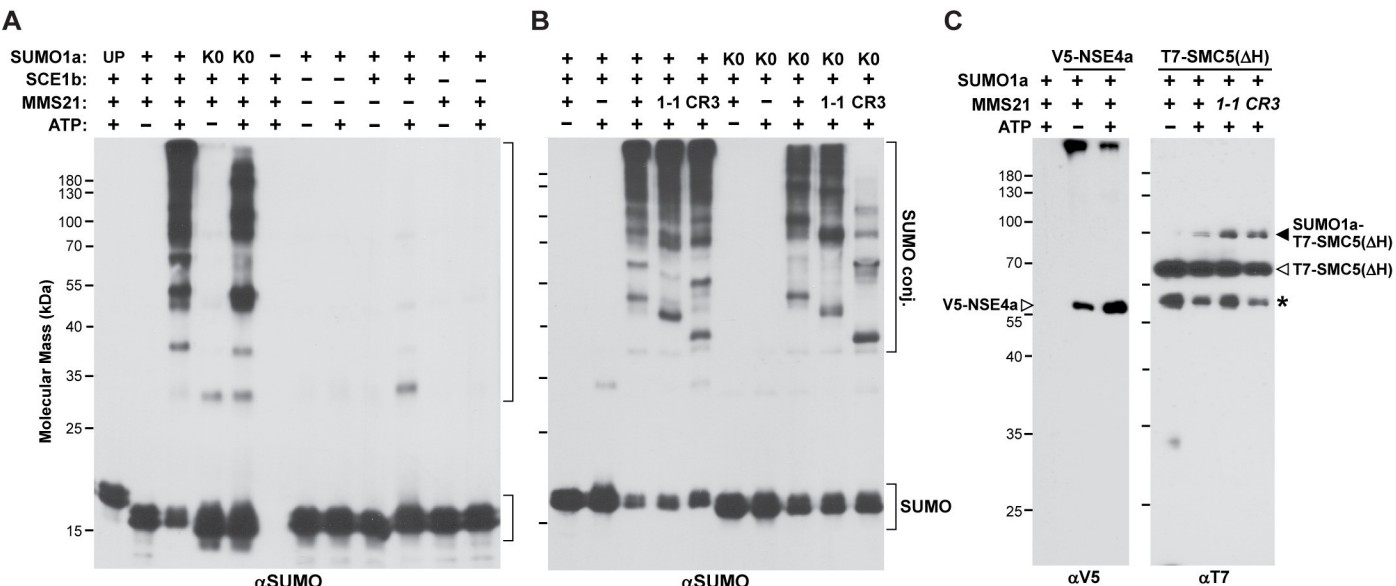

**Fig 10. MMS21 has SUMO Ligase Activity *in vitro*.** Recombinant versions of full-length or mutant forms of MMS21, the heterodimeric SUMO E1 (SAE1/SAE2), the SUMO E2 SCE1b, and either processed (SUMO), unprocessed (UP), or a K0 version of SUMO1a were mixed together in various combinations and incubated at 25°C with or without 5 mM ATP. After quenching the reactions with SDS-PAGE sample buffer, the reactions were subjected to immunoblot analysis with anti-SUMO, anti-V5, or anti-T7 antibodies. Brackets show free SUMO1a and SUMO1a conjugates. **(A)** MMS21 directs SUMO conjugation in complete overnight reactions containing processed SUMO1a, the SAE1/SAE2 E1 heterodimer, the SCE1b E2, and ATP. A silver-stained gel of the reaction mixture is presented in S12A Fig. **(B)** The mms21-1 (1–1) mutant protein and a truncation missing the N-terminal region of MMS21(CR3) encompassing residues Met1-Ala68, retained SUMO ligase activity *in vitro* similar to wild-type MMS21. Reaction were for 1.5 hr as in panel (A). A silver-stained gel of the reaction mixture is presented in S12B Fig. **(C)** MMS21 SUMOylates the ΔHead fragment of SMC5 but not NSE4a *in vitro*. The T7-ΔHead SMC5 polypeptide and full-length V5-NSE4a were added to complete SUMOylation reactions containing wild-type MMS21 or the 1–1 and CR3 variants, and incubated overnight at 25°C with or without 5 mM ATP. The reactions were subjected to SDS-PAGE and immunoblotted with anti-T7 antibodies or anti-V5 antibodies. V5-NSE4a is located by the open arrowhead (left panel). T7-SMC5 (ΔHead) and its SUMOylated form are located by the open and closed arrowheads, respectively (right panel). The asterisk identifies an unknown species that is recognized by the anti-T7 antibodies.

The differing banding patterns for reactions containing wild-type MMS21 and the two mutants suggested that at least some of the SUMOylation by MMS21 reflected self-modification or direct transfer of SUMO1a from the E2 to MMS21. To confirm that MMS21 SUMOylates other proteins, we tested a V5-tagged version of MMS21 *in vitro* using both anti-SUMO1a and anti-V5 antibodies for conjugate detection. While modest SUMOylation of V5-MMS21 was seen in complete reactions using the anti-V5 antibodies, a strong and distinctive smear of conjugates was seen even at low E3 concentrations when using the anti-SUMO1a antibodies, implying that most conjugates were not MMS21-SUMO adducts (S14A–S14C Fig).

Our subsequent attempts to demonstrate that MMS21 will conjugate either NSE4a or SMC5 *in vitro* proved challenging due to the complicated architecture of the SMC5/6 complex that assembles around DNA into a topologically-closed, heterodimeric configuration (Garcia-Rodriquez et al., 2016; S14D Fig). This unique design is particularly evident for SMC5 and SMC6, which use their N-terminal and C-terminal regions to generate a folded Head contact region, and two internal coiled-coil regions that associate to generate a dimeric Arm, which itself is connected by an internal Hinge. The Arm is responsible for binding MMS21 at its N-terminus [49]. Unfortunately, we failed to recombinantly express full-length versions or several partial fragments of either protein without aggregation. However, using a column refolding strategy (see Materials and Methods), we successfully resolubilize full-length NSE4a and a 62-kDa ΔHead fragment of SMC5 missing both sections that comprise the Head region.

When tested in *in vitro* reactions, only the ΔHead fragment of SMC5 was successfully SUMOylated by MMS21. When added to complete reactions also containing processed SUMO1a, MMS21 modified the T7-tagged ΔHead fragment in the presence of ATP (Fig 10C). An adduct was detected at ~100 kDa, which was consistent with the addition of a single SUMO moiety to the 80-kDa ΔHead polypeptide. As with general SUMOylation, both the mms21-1 and mms21-CR3 mutant proteins also directed SUMOylation of the T7-ΔHead fragment. By contrast, equivalent reactions containing full-length NSE4a tagged with V5 failed to generate even after prolonged incubations any new species with apparent molecular masses above the unmodified protein at 62 kDa (Fig 10C). As the Head domains of SMC5 and SMC6 link NSE4 to the rest of the SMC5/6 complex (S14D Fig), this failure likely reflected a disrupted connection between MMS21 and NSE4.

## Discussion

Despite the crucial importance of SUMOylation to plant growth, development, and defense against environmental challenges, the roles of the ligases that drive this post-translational modification and the identit(ies) of their targets remain largely obscure [1,5,50]. Here, we addressed these issues through genetic and molecular analyses of the SUMO E3 MMS21 using maize as a tractable model. Consistent with previous studies with its Arabidopsis MMS21 (HPY2) ortholog, we found through analysis of *Mutator* and CRISPR/Cas9 mutants of varying strengths, that MMS21 has a plethora of functions in maize. Included are roles during gametogenesis, root and shoot growth, and seed development that are likely underpinned by altered nuclear functions, including defects in DNA-damage repair, chromatin dynamics, and proper transcriptional maintenance of proteome balance. For the most part, strikingly similar phenotypes were observed for the maize collection of *mms21* alleles as compared to Arabidopsis *mms21* mutants, including substantially attenuated fertility and seed formation and compromised vegetative development [12,28,29].

Given the phenotypic severity of strong *mms21* mutants, we focused our studies on weaker UniformMu-insertion alleles (*mms21-1/2*) that not only permitted the analysis of reproduction

but also allowed the study of MMS21 throughout the maize life cycle. Particularly notable for these alleles were defects seen in anther maturation and pollen germination, and a delay in seed development that generated poorly-filled, shrunken seeds. The pollen defects could be related to a role for MMS21 in maintaining haploid ploidy levels in male gametes as recently described for Arabidopsis *mms21* mutants [29]. While bulk analysis of SUMO conjugates assembled *in planta* revealed little change in SUMOylation in *mms21* tissues, we found dramatic changes in the transcriptome, with a substantial percentage of mRNAs showing altered abundance in the *mms21-1/2* backgrounds. Instead of finding specific categories of maize genes that were either significantly up- or downregulated, numerous loci were impacted from a wide range of GO categories, implying that MMS21 through its SUMOylation activity, impacts a large swath of the maize transcription. This transcriptional misregulation then translated into an imbalance of the maize proteome, with approximately 24% of the proteins displaying altered accumulation of greater or less than 2 fold even in these mild alleles.

In addition, substantially altered expression in the *mms21-1/2* backgrounds was seen for several potentially informative loci. Included were mRNAs encoding the RNRS2-2 and RNRS-C subunits of the RNR complex important for ribonucleotide metabolism [40], MRE11b needed to maintain genome integrity [37], MORC6 that helps direct DNA silencing [38,39], and the NSE1 and NSE4a subunits of the SMC5/6 complex. All are intimately connected to DNA repair and chromatin dynamics, implying that MMS21-directed SUMOylation modifies these processes in maize which then strongly impacts transcription. Along with the cohesion and condensin complexes, the SMC5/6 complex is especially critical for providing compaction and elasticity to chromatin and the interconversion between euchromatic and heterochromatic states that globally influence gene expression and genome stability [32].

In agreement with a potential broad impact on maize mRNA abundance, our transcriptome profiling surprisingly revealed the ectopic accumulation of zein storage protein mRNAs in *mms21-1/2* shoots. In fact, a large collection of zein mRNAs became the most highly upregulated set of transcripts in both mutant backgrounds, which was also confirmed by qRT-PCR analysis of our *mms21-CR* lines. A wealth of literature has shown that these mRNAs are expressed to high levels in endosperm (and likely embryos) as seeds develop with their encoded zein proteins then accumulating as dense aggregates in specialized protein bodies as seeds mature [41]; these proteins are ultimately consumed during germination to nourish that developing seedling. A similar upregulation was not seen in embryos and endosperm, presumably because these tissues already express zein transcripts to high levels. Also surprising was that this increase in zein mRNA abundances did not coincide with detectable levels of α- and γ-zein proteins in leaves, suggesting either that maize shoots were unable to translate the zein mRNAs out of context, or that shoots were not equipped to stably accumulate zein proteins without the concomitant assembly of appropriate storage compartments.

Collectively, both pervasively altered proteome/transcriptome balance and this unprecedented accumulation of zein mRNAs in *mms21* shoots illustrate a global alteration in gene expression without MMS21. While the exact mechanism(s) remain unclear, we note that SUMOylation in yeast and mammalian cells has been intimately connected to transcriptional repression [51] and the maintenance of heterochromatin stability [52,53]. Moreover, the SUMO E2 and SUMO itself can often be directly mapped to repressed genes, suggesting that this mark is critical for enforcing heterochromatic, transcriptionally-repressed states [54–56]. Consequently, we imagine that the loss of MMS21-directed SUMOylation in the maize *mms21-1/2* shoots causes a loss of heterochromatic organization around genes which are designed to be highly expressed but only under specific developmental conditions. This dysregulation, in turn, derepresses expression in other developmental contexts, thus allowing ectopic expression of genes such as zeins in non-appropriate tissues, or suppressing normally active genes. In support,

we found that some highly up- or downregulated genes (FC ≥16 or -$\log_{10}$ *p*-value ≥20) in the *mms21-2* backgrounds (including those encoding zeins) were clustered within maize chromosomes (S15 Fig and S3 Table). These clusters could reflect heterochromatic regions under the global influence of MMS21-directed SUMOylation. Clearly, further structural analysis of chromatin surrounding zein and other MMS21-impacted genes and mapping chromatin regions that bear SUMO marks in different maize tissues should help clarify this possibility.

Beyond their influence on transcription globally, the *mms21* mutations also impacted DNA damage repair, which is consistent with the role of MMS21 during this process in Arabidopsis and other organisms [13,18,57]. Of the many maize loci encoding factors connected to DNA repair, most had altered expression patterns in the *mms21-1/2* backgrounds, with some upregulated and some downregulated. In accord, we found that *mms21-1/2* roots were hypersensitive to a subset of DNA-damaging agents and housed nuclear DNA with increased DNA breaks as seen by comet assays. Surprisingly, while Arabidopsis *mms21/hpy2* mutants displayed increased endoreduplication (Ishida et al., 2009), suggestive of defects in the endocycle, we did not observe a similar effect with the maize *mms21-1* mutant for leaves, embryos and endosperm tissues, the latter of which is known to undergo numerous scheduled rounds of endoreduplication as seeds mature [46].

Central to understanding MMS21 is the identit(ies) of its substrates, which is complicated by the sheer size of the plant SUMOylome [3–6], and by the fact that SUMOylation can often impact multiple proteins in proximity without direct interaction via a "SUMO spray" mechanism [58]. Global proteomics on the Arabidopsis *mms21-1* mutant implied that few proteins are actual MMS21 targets [5], which is supported in Arabidopsis [5,12] and maize (this report) by the similar profiles of SUMO conjugates seen by immunoblot analysis of total *mms21* and wild-type lysates.

Previous studies with Arabidopsis implicated the SWI/SNF chromatin-remodeling complex subunit BRAHMA [30] and the cell-cycle check point protein DPa, [31] as MMS21 substrates. Here, we discovered through interaction and expression studies, a connection between maize MMS21 and the SMC5 and NSE4a subunits of the SMC5/6 complex involved in chromatin dynamics and DNA repair in Arabidopsis and other organisms [32,59,60]. As predicted, our BiFC studies showed that these interactions between SMC5/NSE4a and MMS21 occur in the nucleus, in agreement with the known location of the SMC5/6 complex [59,60]. NSE4a was previously suggested as an MMS21 substrate based on global SUMOylome profiling with Arabidopsis [5], while SMC5 had not yet been classified as a SUMO substrate in plants. Outside of plants, SMC5, NSE3, and NSE4 have all been shown to be MMS21 substrates, possibly via a SUMO spray mechanism [61,62].

Interestingly, analysis of our panel of UniformMu and CRISPR/Cas9 *mms21* mutants suggested that both the interactions between SMC5 and MMS21 and the phenotypic effects of *mms21* mutants are influenced by subtle alterations in the MMS21 protein sequence, especially near the N-terminal end where the single amino acid substitution in *mms21-CR6* was found to be deleterious. The importance of this region is unclear, but it could harbor a contact used by MMS21 to associate with the rest of the SMC5/6 complex or other influential targets [49]. Interestingly, recent studies by Hays et al. [63] found a similar sensitivity of yeast MMS21 function by single amino acid changes near the N-terminus. While we do not yet know the *in vivo* SUMOylation status of SMC5 and NSE4a in the *mms21* lines, we could show that MMS21 will selectively SUMOylate a partial fragment of SMC5 *in vitro*. A challenge in working with the SMC5/6 complex is its unusual topology which hinders *in vitro* approaches (S14D Fig), and the lack of genetic and molecular tools to easily dissect the process in maize. Work is now underway to overcome these hurdles. As SMC5 protein levels are lower in strong *mms21* mutants, it is possible that SUMO addition by MMS21 protects it from turnover.

In conclusion, our genetic, phenotypic, molecular, transcriptome, and proteome studies identify MMS21 as an essential SUMO ligase for normal maize development. While the exact mechanism(s) remain unclear, we propose that this E3 helps maintain genome integrity against DNA damage and influences chromatin organization to allow faithful transcription throughout maize development, possibly by controlling chromatin compaction through modification of regulators such as BRAHMA, DPa, and subunits of the SMC5/6 complex. Consequently, MMS21-directed SUMOylation is likely key to DNA-damage repair and the heterochromatin/euchromatin transitions necessary for maintaining properly activated/repressed gene states.

## Materials and methods

### Description of *mms21* mutants

The *mms21-1* (mu1068022) and *mms21-2* alleles in the W22 background were found within the UniformMu population [33], and were obtained from the Maize Genetics Cooperation Stock Center or provided directly from the University of Florida collection, respectively. The *Mu* insertion sites were verified by genomic PCR using insertion-specific primers together with the UniformMu border primer, TIR6. The mutant cDNAs were cloned directly from the homozygous lines and sequenced for further analysis. Descriptions of all oligonucleotides are listed in S1 Table.

To generate the CRISPR/Cas9 lines (*mms21-CR1 to 7*) in the Hi-II background, guide RNA-binding sites 1 and 2 (GGACGCGCAAGCCCTAGTCG and GAGGGACGGAAAG TCCGATA) immediately upstream of a PAM were selected using the CRISPR Genome Analysis Tool (http://cbc.gdcb.iastate.edu/cgat/; [64] and the CRISPR RGEN Cas-Designer algorithm (http://www.rgenome.net/cas-designer/; [65,66], followed by secondary screens using MaizeGDB BLAST to avoid regions that might induce off-target effects. The final gRNA oligonucleotides were engineered with 5' overhangs containing TGTT and GTGT sequences for the sense strands of Guide 1 and Guide 2, respectively, and an AAAC sequence was appended to the 5' of all antisense oligonucleotides. Guide 1 oligonucleotides were annealed to one another and phosphorylated with T4 polynucleotide kinase before ligation into the BtgZ1-digested pENTR-gRNA1 [67], thereby placing their expression under the control of the rice RNA polymerase-III *pU6.1* promoter. The resulting plasmid was digested with *Bsa*I and ligated to phosphorylated Guide 2 oligonucleotides to generate the pENTR-gRNA1-*ZmMms21* plasmid that would drive expression from another rice *U6* gene promoter (*pU6.2*). LR clonase (Invitrogen) reactions recombined the guide RNA cassette into the pGW-Cas9 [68] gateway cassette to generate the final pGW-Cas9-gRNA-*ZmMms21* construction, which was subsequently introduced into *Agrobacterium tumefaciens* strain EHA101. This plasmid was transformed into the *Z. mays* Hi-II line by the Plant Transformation Facility at Iowa State University (https://www.biotech.iastate.edu/biotechnology-service-facilities/plant-transformation-facility). The nature of the *mms21-CR* mutations was determined by PCR amplification of the genomic region encompassing the guide RNA target sites, followed by Sanger sequencing of ExoSAP (Applied Biosystems)-treated PCR amplicons to identify the exact lesions.

### Plant growth and phenotypic analyses

The *mms21-1* and *mms21-2* mutants were backcrossed five times to the W22 ACR inbred line [33] before a self-cross to obtain homozygous plants, whereas the *mms21-CR* alleles were back-crossed twice to B73 and self-pollinated to generate segregating populations. The plants were grown to maturity using 3:1 Metromix 900 Turface MVP (Sungrow) in the greenhouse under a 16-hr light/8-hr dark photoperiod with 1000 W metal-halide lamps used to

supplement natural light, and 24–27°C and 22–25°C day and night temperatures, respectively. Watering was performed as necessary with Peters 15-5-15 Cal-Mag special supplemented with Sprint 330 iron chelate until flowering, at which point watering continued without amendments [24].

Heights were measured from the soil level to the youngest ligule beginning 14 DAS. Leaf length was measured from the auricle to the tip, and width was measured on fully mature plants at the halfway point of the fourth, sixth, or eighth leaves from the tassel. For fresh and dry weight measurements of roots and shoots, seeds were sown into ~1.5 cm fine sand Turface Quick Dry layered atop coarse Turface MVP. Plants were watered twice daily; every fifth watering day used modified Hoagland solution [24]. To correct for delays in germination, tissues were harvested 7 or 14 d after first emergence of the coleoptile from the Turface growth substrate. For analysis of shoot and root growth, seedlings at 7 or 14 d after emergence were separated from Turface in a water bath and briefly dried with paper towels, and then roots were scanned and fresh weights were measured. Dry weights were measured after overnight drying at 55°C. Root area was determined in ImageJ from scanned images by selecting the blue channel after RGB separation, inverting the lookup table, thresholding by MaxEntropy, and then measuring the area of the resulting image.

For the heat stress assays, plants were grown in controlled environment chambers (Conviron) at 28°C under 16-hr light/8-hr dark photoperiod on soil containing 3:1 Metromix:Turface. Seedlings at the 3-leaf stage were transferred to a pre-equilibrated chamber set at 42°C and 60% humidity; after 30 min, tissue was collected and immediately flash frozen to liquid nitrogen temperatures.

## Pollen germination assays

Pollen was collected from flowering tassels from which the florets and anthers were gently removed. Pollens were incubated in solid germination medium containing 15% sucrose, 0.6% bacto-agar, 0.03% calcium nitrate, and 0.01% boric acid [69] for 3 hr at 28°C in darkness. Germination was scored by protrusion of the pollen tubes seen microscopically.

## Immunoblot analyses

Proteins separated by SDS-PAGE were electrophoretically transferred onto Immobilon-P polyvinylidene difluoride (PVDF) membranes (Millipore), which were then blocked at room temperature with 5% nonfat dry milk powder and 0.2% Tween-20 dissolved in phosphate-buffered saline (PBS) containing 154 mM NaCl and 10 mM $Na_2PO_4$ (pH 7.4). The membranes were incubated for 2 hr at room temperature or overnight at 4°C with primary antibodies at a 1:10,000 dilution in blocking solution, and washed with PBS containing 0.2% Tween-20. The membranes were then incubated for 2 hr with horseradish peroxidase (HRP)-decorated goat anti-rabbit secondary antibodies (SeraCare, Cat. No. 5220–0341) at 1:10,000 dilution in blocking solution, followed by washes with PBS containing 0.2% Tween-20. Chemiluminescence was generated using the SuperSignal West Pico Plus Chemiluminescent Substrate (Thermo Fisher) and captured with X-ray film. Primary polyclonal antibodies included those against AtSUMO1 [20], histone H3 (Abcam; ab1791), α and γ zeins [41], the T7 Tag polyclonal antibody coupled to HRP (Invtrogen; Cat. No. PA1-33133), and the V5 tag (Invitrogen; Cat. No. PA1-993).

## RNA isolation, qRT-PCR, and cDNA cloning

Total RNA was isolated from 10-DAS shoots using the RNeasy Plant Mini Kit (Qiagen). RNA integrity was assessed after treatment with DNase I (ThermoFisher) by agarose gel

electrophoresis and OD260/OD280 measurements. For RT-PCR, one μg of total RNA was subjected to the Superscript III First Strand Synthesis System (ThermoFisher) using an oligo $(dT)_{20}$ primer, followed by qRT-PCR using the Light Cycler 480 and SYBR Green I master mix (Roche Diagnostics) in combination with gene-specific primers (S1 Table). Relative transcript abundances were calculated by the $2^{-\Delta\Delta CT}$ method [70], using amplification of the maize *ACT1* gene as a reference. For Y2H, BiFC, and recombinant protein expression, open reading frames were PCR amplified from cDNA using Phusion High-Fidelity DNA Polymerase (Thermo-Fisher) and then recombined into pDONR221 via the Gateway BP clonase II reaction (ThermoFisher). All clones were sequenced in their entirety to confirm the absence of secondary mutations.

## RNA-seq analysis

Shoots were harvested 10 DAS from the *mms21-1 and mms21-2* plants and their corresponding W22 inbred line grown in a 25°C growth chamber, and immediately frozen in liquid nitrogen and stored in -80°C. Endosperm and embryos were dissected from seeds at 16 DAP and frozen in liquid nitrogen. The *mms21-2* seeds were identified visually at the time of harvesting, while the *mms21-1* seeds were scored by genomic PCR after dissection of the cobs. Total RNA was extracted from three biological replicates, each of which was prepared from three seeds from the same segregating ear. Shoot and embryo RNAs were isolated using RNeasy Plant Mini Kit (Qiagen), while the endosperm RNAs were first enriched using TRIzol (Invitrogen) before isolation.

To prepare for RNA-seq analysis, the RNA samples were digested with DNase I at 37°C for 10 min to remove contaminant DNA. RNA-seq library construction and sequencing were performed by BGI Americas Inc. (Hong Kong, China), using their DNBseq technology platform (PE150, 20 million reads). Roughly 7-Gb of raw data generated from each sample, after removing reads from the sequencing adaptors, were further screened to remove low-quality sequences using Trimmomatic v0.39 under the PE mode [71], which truncated the reads for base quality <15 within 4-base windows and kept only reads longer than 36 bases after trimming under the parameters: phred33, LEADING:3, TRAILING:3, SLIDINGWINDOW:4:15, and MINLEN:36). Trimmed reads were uniquely aligned to the B73 RefGen_v4.48 maize (*Zea mays*) genome assembly using STAR [72]. Counts for gene expression were obtained using HTSeq [73].

DEG analysis was conducted with the DESeq2 Bioconductor package version 1.28.1 in R 4.0, using the total uniquely mapped read count per gene parameter as input [74]. Up- or downregulated DEGs were selected from both *mms21-1* and *mms21-2* alleles based on adjusted $p$-values <0.05 and FCs $\geq$2, and were used as inputs to draw the Venn diagrams in R. PCA was calculated in R to visualize clustering of the datasets. Correlation analysis were performed using the $log_2$-FC values provided by DESeq2, filtered using an adjusted $p$-value <0.05, and drawn with the R packages—ggpubr and ggplot2 [75,76]. Heatmaps were draw with pheatmap in the R package [77], based on the same datasets used for the correlation analyses. For gene set enrichment analysis (GSEA), all genes identified from RNA-Seq were first filtered to remove those that were not significant based on adjusted $p$-values <0.05. Filtered genes were ranked based on their calculated value using equation: $-log_{10}(p\text{-value}) * sign(log_2\text{-FC})$ and imputed into the clusterProfiler package for GSEA in R [78]. GSEA was used to assess whether genes in biologically predefined sets occurred toward the top or bottom of a ranked list of all examined genes more than expected by chance. Significant gene sets were called as those with an adjusted $p$-value < 0.05 [79]. Volcano plots were generated in ggplot2 [76]. Mapping of DEGs on the maize chromosomes were determined by TBtools [80].

## Proteomic profiling

Total maize proteome profiling was performed as described by McLoughlin et al. (2018) with minor modifications. Proteins were extracted into 50 mM HEPES (pH 7.5), 5 mM $Na_2EDTA$, 2 mM dithiothreitol and 1 × plant protease inhibitor cocktail (Sigma-Aldrich), and precipitated with methanol-chloroform. Pellets were resuspended into 8 M urea, reduced with 20 mM dithiothreitol, and alkylated with 20 mM iodoacetamide for 1 h before digestion with 0.5 µg trypsin (Promega) overnight. The resulting peptides were desalted and concentrated with a 100-µl Bond Elut OMIX C18 pipette tip (Agilent Technologies) and resuspended in 20 µl 5% acetonitrile and 0.1% formic acid. The peptides were then analyzed with a Q-Exactive Plus mass spectrometer (Thermo Fisher Scientific) after reversed-phase nano-HPLC separation with a 25-cm analytical C18 resin column (Acclaim PepMap RSLC; Thermo Fisher Scientific) and a 5 to 95% acetonitrile gradient in 0.1% formic acid (FA) at a flow rate of 250 nL/min for 135 min. The mass spectrometer was operated in the data-dependent mode to automatically switch between full-scan MS and MS/MS acquisition. Data-dependent acquisitions were obtained using Xcalibur 4.0 software in positive-ion mode. MS1 spectra were measured at a resolution of 70,000 with an automatic gain control of $1 \times 10^6$, a maximum ion time of 50 msec, and a mass range of 300–1,800 m/z. Up to 12 MS2 scans, with a charge state of 2 to 4, were triggered at a resolution of 17,500, an automatic gain control of $5 \times 10^5$ with a maximum ion time of 120 msec, a 1.6-m/z isolation window, and a normalized collision energy of 28. MS1 scans that triggered MS2 scans were dynamically excluded for 30 sec. Each biological replicate was analyzed by two technical replicates.

The resulting MS data sets were searched against the maize B73 proteome database (Zm-B73-REFERENCE-GRAMENE-4.0 from www.maizegdb.org) using Proteome Discoverer (version 2.5.0.400; Thermo Fisher Scientific) and a list of common protein contaminants [42]. Peptides were assigned by SEQUEST HT, allowing a maximum of two missed tryptic cleavages, a minimum peptide length of 6, a precursor mass tolerance of 10 ppm, and fragment mass tolerances of 0.02 Da. Carbamidomethylation of Cys and oxidation of Met were specified as static and dynamic modifications, respectively. Protein abundances reflected the average of two technical replicates if proteins were detected twice or used directly if the proteins were only detected in one technical replicate. Values among samples were normalized using the average values of 150 proteins considered least variable among the samples (using SD/average as ranking [42]). Only those proteins found in at least 2 biological replicates were included in the final datasets. Missing values for proteins not identified in all replicates were imputed in R using random draws from a Gaussian distribution centered around the minimal 1% of the observed values in that sample. To assess statistically significance differences among samples, the Limma algorithm in the "DEP" R package [81] was applied with the following filters: $\log_2$ FC >1 (FC >2 on a linear scale) and adjusted $p$-value (corrected with Benjamini-Hochberg method) <0.05.

## Genotoxic stress assays

Maize seeds were surface sterilized with 15% bleach containing 0.01% Triton X-100 for 15 min with shaking and then washed extensively with water. Sterilized seeds were first germinated on wet sterile filter paper for 3 d at room temperature and then transplanted into test tubes filled with Murashige and Skoog growth medium also containing 3% sucrose and 1% agar, which was supplemented with methyl methanesulfonate, mitomycin C, hydroxyurea, bleocin, or zebularine (Sigma-Aldrich). *mms21-1* and *mms21-2* seeds were germinated 2 d ahead of the W22 control to compensate for their slower germination. Root lengths were measured after 1 week of growth; lengths for the drug-treated samples were expressed relative to those observed under control conditions.

## Comet assays

Comet assays were performed as described [45] with some modifications. Briefly, young roots were chopped with a razor blade in 400 mM Tris-HCl (pH 7.5). The nuclear suspensions were filtered through 40 μm cell strainers (Corning) to remove cell debris, mixed with an equal volume of 1% low melting point agarose prepared with PBS, and then spread on microscope slides cooled to 4˚C for 5 min to solidify the mixture. Embedded nuclear DNA was unwound at 4˚C for 5 min in electrophoresis buffer (1 mM $Na_2EDTA$ and 300 mM NaOH (pH> 13)), and then electrophoresed at 300 mA for 20 min at 4˚C. Slides were rinsed 3 times with 400 mM Tris-HCl (pH 7.5) before staining the DNA with 100 μg/ml propidium iodide. Fluorescence was visualized with a Leica SP8 laser scanning confocal microscope equipped with a 40X oil objective, using 488-nm for excitation and emission between 500 and 530 nm.

## DNA flow cytometry analysis

Leaves at 10 DAS and whole seeds, endosperm and embryos dissected at 16 DAP were chopped with fine razor blades directly into ice cold Galbraith's buffer (45 mM $MgCl_2$, 30 mM sodium citrate, 20 mM MOPS (pH 7.0), 0.1% Triton X-100, and 1% PVP-40). Nuclei were enriched by filtration through 30 μm CellTrics nylon filters (Sysmex), digested with RNase A for 10 min, and stained with 50 μg/mL of propidium iodide for 30 min in the dark. Flow cytometry was performed using an Accuri C6 cytometer (BD Biosciences) setup to measure at least 10,000 PI fluorescence events per sample, with emission detected using the FL2 optical filters (585/40 nm), and a minimum threshold intensity of $10^4$ required for data collection.

## Yeast two-hybrid (Y2H) analysis

Y2H assays were performed using the Matchmaker 3 Y2H System (Clontech) as described [82] using maize components. The cDNA sequences of *Mms21* (Zm00001d039007), *Sumo1a* (Zm00001d012042), *Sce1b* (Zm00001d027427), Sce1f (Zm00001d002572), *DPa* (Zm00001d011597), *Nse4a* (Zm00001d036797), *Smc5* (Zm00001d014500), and coding sequence for the N-terminal portion (Met1-Arg1195) of *Brm* (Zm00001d014977) were cloned using RNA isolated from 10-DAS B73 seedlings. Sequence-confirmed clones were introduced into pDONR221 via Gateway BP clonase II reactions (ThermoFisher) in-frame either with that encoding the Gal4 DNA-binding domain in the pGBKT7-GW vector, or the Gal4-activation domain in the pGADT7-GW vector (Lu et al, 2010). Pairwise combinations of coding sequences in pGBKT7-GW and pGADT7-GW (or the empty vectors as controls) were co-transformed into the AH109 yeast strain. Positive clones were selected using minimal synthetic medium (SD medium) minus tryptophan and leucine (SD/-Trp/-Leu). Protein-protein interactions were identified after 2–3 d growth at 28˚C on SD medium lacking adenine, leucine, tryptophan and histidine (SD/-Ade/-His/-Leu/-Trp).

## Bimolecular fluorescence complementation (BiFC)

BiFC assay were accomplished using *N. benthamiana* leaves as described [83]. Sequence-confirmed coding sequences of maize genes, cloned into pDONR221 as above, were recombined in-frame via Gateway LR clonase II reactions with the N- or C-terminal halves of EYFP in the pSITE-N-EYFP-C1 or pSITE-C-EYFP-C1 vectors (ABRC Cat. No. CD3-1648 and CD3-1649), respectively. The resulting plasmids were introduced into *A. tumefaciens* strain GV3101 by freeze-thaw transformation, and positive clones were cultured overnight and resuspended to an $OD_{600}$ = 0.4 in 10 mM MES (pH 5.7), 10 mM $MgCl_2$, and 200 μM acetosyringone. *A. tumefaciens* suspensions were incubated for 4 to 6 hr in darkness at room temperature before

infiltration into 4 to 6-week-old *N. benthamiana* leaves. Leaf sections were excised 40 to 45 hr after infiltration and visualized with a Nikon A1+ confocal laser scanning microscope equipped with a 40x oil objectives (numerical apertures 1.30). Representative fluorescent and brightfield images were captured using Nikon Imaging Software Elements, using 488-nm light for GFP fluorescence excitation and 500–530 nm light for emission capture.

## Recombinant protein expression and purification

Unless otherwise noted, all recombinant proteins were expressed as full-length versions with appended tags. The coding sequences of maize SUMO1a and MMS21, obtained by PCR-amplification of B73 cDNA, were first cloned into pDONR221 as described above. Other maize SUMO components, including processed SUMO1a and the K0-SUMO1a(H89R) variant (both ending in its diGly motif), SCE1b-Myc, HA-SAE1, SAE2a-FLAG bearing the indicated tags were as described [24]. The constructions were further modified to include coding sequence for an N-terminal 6His tag to facilitate protein purifications and further subcloned into pRSFDuet-1 expression vector for expression in the *E. coli* strain BL21(DE3) pLysS (Promega). The cells were cultured at 37˚C to an $OD_{600}$ of 0.4 ~ 0.8 in of LB medium supplemented with 50 µg/mL kanamycin and 34 µg/mL chloramphenicol, followed by a 5-hr induction with 1 mM isopropyl-β-D-thiogalactopyranoside at 37˚C. Cells were harvested by centrifugation at 10,000 X *g* for 5 min at 4˚C, frozen in liquid nitrogen, and lysed in 15–20 mL BugBuster Master Mix (Millipore Sigma). The 6His-tagged proteins were affinity purified by nickel-nitrilo-triacetic acid (Ni-NTA) agarose chromatography (QIAGEN). The bacterial cell lysates, supplemented with one tablet of Pierce Protease Inhibitor Tablets (ThermoFisher Scientific) and 10 mM imidazole, were applied to 5 mL of PBS-washed Ni-NTA beads (Bio-Rad) at 4˚C and incubated for 30 min with mixing. The Ni-NTA beads were washed once with 30 mL of $NaH_2PO_4$ wash buffer (50 mM $NaH_2PO_4$ (pH 7.8), 300 mM NaCl, and 20 mM imidazole), and once with 30 mL HEPES wash buffer (50 mM HEPES (pH 7.8), 300 mM NaCl, and 40 mM imidazole). Bound proteins were eluted twice with 1 mL of Tris elution buffer (50 mM Tris-HCl (pH 7.8), 100 mM NaCl, and 300 mM imidazole). Eluants were dialyzed at least twice for 3 hr at 4˚C against 1 L of dialysis buffer (40 mM HEPES (pH 7.5), 50 mM NaCl, 10 mM $MgCl_2$, and 10% glycerol) to remove the imidazole.

To purify and solubilize NSE4a and the ΔHead (D291-E783) fragment of SMC5, IPTG-induced *E. coli* cells were first suspended in BugBuster Master Mix. The insoluble material was collected by centrifugation at 10,000 X *g*, solubilized into 20 mM Tris-HCl (pH 8.0), 0.5 M NaCl, 5 mM imidazole, 6 M guanidine-HCl and 1 mM 2-mercaptoethanol, loaded onto a Ni-NTA column equilibrated in the same buffer, and incubated at 4˚C with gentle shaking for 1 hr. The beads were washed with 10 volumes of Wash Solution-1 (20 mM Tris-HCl (pH 8.0), 0.5 M NaCl, 5 mM imidazole, 6 M guanidine-HCl, and 1 mM 2-mercaptoethanol), and 10 volumes of Wash Solution-2 (20 mM Tris-HCl (pH 8.0), 0.5 M NaCl, 20 mM imidazole,1 mM 2-mercaptoethanol, and 6 M urea). Refolding of the bound proteins were performed using a linear 6–0 M urea gradient, starting with the Wash Solution 2 and finishing with Wash Solution 2 minus urea. Bound proteins were eluted with 20 mM Tris-HCl (pH 8.0), 0.5 M NaCl, 500 mM imidazole and 1 mM 2-mercaptoethanol, and dialyzed twice at 4˚C against 1 L of dialysis buffer to remove the imidazole.

## *In vitro* SUMOylation assays

Standard *in vitro* SUMOylation assays were performed with 20 µL reaction mixtures containing 40 mM HEPES (pH 7.5), 10 mM $MgCl_2$, 0.2% Tween-20, 50 mM NaCl, 4 mM dithiothreitol, 4 µg SUMO1, 500 ng SAE1 and 200 ng SAE2a (E1), 400 ng SCE1b (E2), and 1.3 µg

MMS21(E3), with or without 5 mM ATP. Unless otherwise indicated, the reactions were performed at 25°C for 1.5 hr and quenched by adding SDS-PAGE sample buffer (0.25 M Tris-HCl (pH 6.8), 10% (w/v) SDS, 30% (v/v) glycerol, and 0.05% bromophenol blue). The *in vitro* SUMOylation reactions with the ΔHead fragment were performed overnight in the same reaction buffer as above but with 2.6 μg of MMS21 and 1 μg of the ΔHead substrate (as determined by the BCA protein assay kit (ThermoFisher)).

## Accession numbers

The raw RNA-seq files are available at the NCBI Sequence Read Archive database under the submission number PRJNA685214 (https://www.ncbi.nlm.nih.gov/sra/PRJNA685214). The. raw,.msf,.mzid and.mzML files for the MS datasets are available in the ProteomeXchange database under accession number PXD026853 within the PRIDE repository (http://www. proteomexchange.org/). Gene sequence data can be found in the GenBank/EMBL libraries under the following accession numbers: *Zea mays*, *Zm*MMS21(AQK88509.1); *Aquilegia coerulea*, *Ac*MMS21(PIA35101.1); *Oryza sativa*, *Os*MMS21(XP_015640264.1); *Arabidopsis thaliana*, *At*MMS21 (NP_188133.2); *Physcomitrella patens*, *Pp*MMS21(XP_001767320.1); *Glycine max*, *Gm*MMS21 (NP_001241980.2); *Selaginella mollendorffii*, *Sm*MMS21(XP_024515397.1); *Setaria italica*, *Si*MMS21 (XP_004961220.1); *Saccharomyces cerevisiae*, *Sc*MMS21 (GFP73070.1); *Homo sapiens*, *Hs*NSE2(NP_001336414.1).

## Supporting information

**S1 Fig. Maize *mms21* Mutants Dampen Vegetative Growth.** Shown are measurements of tissues from homozygous *mms21-1*, *mms21-2*, and W22 plants. **(A)** Quantification of fresh weight, dry weight, and root area of plants harvested at 7- and 14 (Days After Germination) DAG. Each bar represents the mean of 3 biological replicates, each with at least 3 plants (±SE). **(B)** Quantification of leaf length and width for the 4[th], 6[th], and 8[th] leaves (counted from the tassel) at maturity. Each bar represents the mean of 3 biological replicates, each with at least 3 plants (±SE). The a, b, and c notations identify values that were significantly different from one another and the normal sibling control, as determined by one-way ANOVA followed by the Tukey's post hoc test (*p*-value <0.05). **(C)** Representative epidermal surface impressions of leaves. Micrographs of glue slide impressions were prepared from abaxial surface of mature leaves adjacent to the tassel. Bars = 100 μm. **(D)** Box plots of cell lengths for leaves grown as in panel (C). Cell lengths were measured using ImageJ and plotted in R (n = 200).
(TIF)

**S2 Fig. Maize *mms21* Mutants Impact Anther Development. (A)** Quantification of defects seen for *mms21-1* and *mms21-2* anther development as compared to those from normal siblings. Traits measured include: days to tassel emergence, days to silking, and days to pollen shedding. Each bar represents the average of three biological replicates (±SD). **(B)** Dissection of male florets showing that male floral morphology is relatively normal for *mms21-1* but abnormal for *mms21-2*. Scale bar = 8 mm.
(TIF)

**S3 Fig. Maize *mms21* Mutants Compromise Seed Development.** Shown are cobs from self-pollinated W22, and *mms21-1/+*, and *mms21-2/+* plants from 8 DAP to maturity. Arrowheads locate obvious defective kernels.
(TIF)

**S4 Fig. The Collection of CRISPR/Cas-9 Alleles are Phenotypically Similar to Those Generated by *Mu* Transposition. (A)** Mature ears from self-pollinated heterozygous *mms21*

CRISPR/Cas9 (*CR*) alleles showing the appearance of defective kernels (arrowheads). Scale bars = 1 cm. **(B)** Abnormal morphology of *mms21-CR* seeds. The abgerminal (top), germinal sides (middle) and the saggital sections (bottom) of one representative seed is shown. Scale bar = 5 mm. **(C)** Homozygous *mms21-CR* mutants imaged 2-week after planting. Scale bars = 5 cm. **(D)** *mms21-CR4* and *mms21-CR7* plants imaged at flowering. Scale bars = 30 cm. Each seed/plant was compared to its phenotypically normal sibling.
(TIF)

**S5 Fig. The *mms21* Transcriptomic Data Showed Strong Consistency When Comparing Biological Replicates. (A)** The scatter plots comparing the transcript profiles obtained from three biological replicates from either W22, *mms21-1*, or *mms21-2* shoots, developing embryos, or endosperm. Transcript abundances acquired by RNA-seq were expressed as $\log_2$-transformed normalized values. The total number of transcripts analyzed is shown at the bottom right corner. The dashed lines show a correlation = 1, correlation of the first two biological replicates were plotted, while the $\log_2$-transformed transcript abundances for the third biological replicate are superposed onto the scatter plot by the color gradient shown on the right. **(B)** The Pearson correlation coefficients among the samples for each genotype.
(TIF)

**S6 Fig. GO Term and KEGG Enrichment Analyses of the *mms21* Transcriptomic Datasets. (A-C)** GO term enrichment analysis of DEGs in the *mms21-2* versus W22 shoots. The vertical coordinates indicate the enriched GO terms, and the horizontal coordinates show the normalized enriched score for each GO term. Negative values indicate downregulation, positive values indicate upregulation. GO enrichment was performed using all the three sub-ontologies: Biological Process **(A)**, Molecular Function **(B)**, and Cellular Component **(C)**. **(D)** KEGG enrichment analysis of the DEGs. The vertical coordinates are the enriched pathways, and the horizontal coordinates are the normalized enriched score for each GO term.
(TIF)

**S7 Fig. Identification of *mms21*-Induced DEGs by Volcano Plots.** Volcano plot representation of DEGs in *mms21 versus* the W22 based on FC and $-\log_{10}$ adjusted *p*-values. Shown are transcriptome analyses of *mms21-1* shoots **(A)**, *mms21-1* embryos **(B)**, *mms21-1* endosperm **(C)**, and *mms21-2* endosperm **(D)**. Analysis of other tissues are shown in Fig 6A. The horizontal and vertical dashed lines indicate a FC = 2 and an adjusted *p*-value = 0.05. Transcripts encoding proteins involved in DNA repair, maize zein storage proteins, and the SUMO pathway are indicated in red, green and blue points, respectively. All other transcripts are shown in grey. Specific mRNAs of interest that were significantly altered in expression are noted.
(TIF)

**S8 Fig. Maize *mms21* Shoots do not Accumulate Detectable Levels of Zein Proteins.** Ethanol soluble proteins from *mms21-1*, *mms21-2*, and W22 shoots harvested at 10 DAS were subjected to SDS-PAGE and immunoblot analysis with antibodies against the 19- and 22-kDa α-zeins, or the 27-kDa γ-zeins. Five-fold serial dilutions after a 10-fold dilution of W22 endosperm dissected from dry seeds are shown in the left lanes for comparisons. Seedling samples were loaded on equal dry weight basis. The migration positions of the zeins are indicated by the arrowheads.
(TIF)

**S9 Fig. Comparative Proteomic Analysis of the *mms21-1* Mutant and Normal *mms21* Siblings. (A)** Altered proteome profile for the *mms21-1* mutant. The volcano plot depicts protein abundance changes for 4,171 proteins detected by MS from *mms21-2* leaves as compared to those of its normal sibling. **(B)** Proteome profile comparison of leaves from *mms21-1* normal

siblings and *mms21-2* normal siblings. Each dot represents one protein that had detectable expression in both samples and was plotted based on its $\log_2$ FC in abundance (mutant/normal siblings) and its $-\log_{10}$ *p*-value of significance based on the three biological replicates, each with two technical replicates. The horizontal and vertical dashed lines mark a FC = 2 in protein abundance and a *p*-value = 0.05, respectively. Histone proteins used to confirm data normalization are shown as green. SUMO pathway components and DNA repair-associated proteins are highlight in blue and red, respectively.
(TIF)

**S10 Fig. The *mms21* Mutations do not Substantially Impact the Ploidy Level of Maize.** Nuclei were isolated from seedling leaves, whole seeds, endosperm and embryos, stained with propidium iodine, and subjected to FACS sorting to determine ploidy number based on fluorescence counts. **(A)** Distribution of nuclei ploidy levels in seedling leaves from 10 DAS plants (2n and 4n). **(B)** Quantification of ploidy levels derived from seedling leaf nuclei analyzed in panel (A). Each bar represents the percent of the respective average totals obtained from three biological replicates, each of which was prepared from one seedling (±SD). **(C)** Distribution of nuclei at various ploidy levels (2n to 96n) in whole seeds, and dissected endosperm and embryos at 16 DAP. The predicted ploidy levels of the various fluorescence peaks are indicated. **(D)** Quantification of ploidy levels embryo and endosperm nuclei analyzed in panel (C). Each bar represents the percent of the respective average totals obtained from three biological replicates, each of which was prepared from three pooled seeds (±SD).
(TIF)

**S11 Fig. Controls for the BiFC Assays.** Pairwise expression of MMS21 and its potential inter-actors fused to the N-terminal (nYFP) or C-terminal (cYFP) halves together with the nYFP and cYFP fragments by themselves. *N. benthamiana* leaf epidermal cells were co-infiltrated with the indicated plasmid combinations, and fluorescence signals were detected by confocal fluorescence microscopy 40–45 h after infiltration. Shown are the fluorescence images alone or merged with their companion bright field images. Only the cYFP-SCE1b construction expressed by itself generated a subtle fluorescence signal due to auto-activation. Scale bars = 40 μm.
(TIF)

**S12 Fig. Protein-Stained Gels for a Collection of *In vitro* SUMOylation Assays Involving MMS21.** The SUMOylation reaction mixtures are identical to those described in Fig 10A and 10B. The mixtures were subjected to SDS-PAGE and stained for protein with silver. The migration position for each component is indicated by the arrowheads. An unknown contaminant is highlighted by the asterisk.
(TIF)

**S13 Fig. Optimization of the Assay Conditions for *In vitro* SUMOylation by Maize MMS21.** Recombinant versions of full-length SUMO E1 (SAE1/SAE2), the SUMO E2 SCE1b, the processed and active version of SUMO1a, and full-length MMS21 were affinity purified and mixed in various combinations with or without ATP. After quenching the reactions with SDS-PAGE sample buffer, the mixtures were subjected to SDS-PAGE and immunoblot analysis with anti-SUMO1 antibodies. **(A)** Time course of SUMO conjugation for reactions containing only SUMO1a, the SAE1/SAE2 E1 heterodimer, the SCE1b E2, and ATP. O/N, overnight. **(B)** Time course of SUMO conjugation for reactions as panel (A) but with 10 times more of SCE1b E2 enzyme. O/N, overnight. **(C)** Time course for MMS21-directed SUMO conjugation at 25˚C for reactions containing SUMO1a, the SAE1/SAE2 E1 heterodimer, the SCE1b E2, and ATP. **(D)** Time course for MMS21-directed SUMO conjugation at 42˚C as in panel (C). **(E)** Complete SUMO conjugation reactions containing SUMO1a, the SAE1/SAE2

E1 heterodimer, the SCE1b E2, MMS21 E3, and various concentrations of ATP. **(F)** Pyrophosphate (PPi) inhibits SUMOylation. MMS21-directed SUMO conjugation was conducted in the presence of increasing concentration of PPi in reactions containing SUMO1a, the SAE1/SAE2 E1 heterodimer, the SCE1b E2, and 2 mM ATP. Unless indicated otherwise, the assays were performed at 25˚C in 20 μL reaction volumes containing 4 μg of SUMO1a, 500 ng of SAE1 and 200 ng of SAE2a (E1), 400 ng of SCE1b (E2), and 1.3 μg of MMS21(E3), with or without 2 mM ATP. The reactions in panels (E) and (F) were performed for 1.5 hr.
(TIF)

**S14 Fig. MMS21 Acts as an Enzyme Catalyzing *in vitro* SUMOylation Reactions. (A and B)** *in vitro* reactions showing that MMS21 mainly SUMOylates other proteins within the reaction mixtures and not only MMS21 itself. Shown are *in vitro* SUMOylation in complete reaction mixtures containing increasing concentrations of V5-tagged MMS21 along with processed and active SUMO1a, the SAE1/SAE2 E1 heterodimer, the SCE1b E2, and 5 mM ATP. Reactions were conducted at 25˚C for 1.5 hr in 20 μL volumes containing 4 μg of SUMO1a, 500 ng of SAE1 and 200 ng of SAE2a (E1), 400 ng of SCE1b (E2), and various amounts of MMS21 (E3), with or without 2 mM ATP as indicated. One part V5-MMS21 equals 1.3 μg. The products were subjected to SDS-PAGE and immunoblot analysis with anti-SUMO1 **(A)** or anti-V5 antibodies **(B)**. **(C)** The N-terminal V5 tag did not impact the SUMO ligase activity of MMS21. Shown are complete SUMO conjugation reactions as in (A) containing 1.3 μg MMS21 expressed with (right panel) or without (left panel) the V5 tag. The arrowhead and bracket locate V5-MMS21 and free SUMO1a, respectively. The asterisks indicate SUMOylated forms of V5-MMS21 generated during the reaction. Note that the profile of conjugates detected with anti-SUMO1 antibodies differ markedly from that detected with anti-V5 antibodies. **(D)** Cartoon of the predicted three-dimensional structure of the SMC5/6 complex bound to DNA and containing its accessory factors NSE1, NSE3, and MMS21 (also known as NSE2). The Hinge, coiled-coil Arm, and Head domains of SMC5 and 6 are indicated. The SMC5/6 complex proteins known to be SUMOylated are shown. Adapted from [47]. S, SUMO.
(TIF)

**S15 Fig. Chromosomal Locations of Genes Differentially Up or Downregulated in *mms21-2* Shoots.** Positions within the 10 maize chromosomes were mapped by TBtools [80] for the collection of 146 DEG showing a FC >16 or <-16 or a -$\log_{10}$ adjusted *p*-value >20 (adjusted *p*-value <$1e^{-20}$), either up or down. Centromeres are shown in red. Genes encoding zeins are highlighted in red while those for MMS21, NSE4a, MOR6, and MRE11b are highlighted in blue.
(TIF)

**S1 Table. Oligonucleotide Primers Used in This Study.**
(XLSX)

**S2 Table. List of DEGs Analyzed in Fig 8A from *mms21* Embryos Involved in DNA Repair.**
(XLSX)

**S3 Table. Gene Identification Numbers for Genes in S15 Fig.**
(XLSX)

## Acknowledgments

We thank Kehui Wang (Washington University in St. Louis) for technical support in phenotypic studies and David Holding (University of Nebraska-Lincoln) for providing the anti-zein antibodies.

## Author Contributions

**Conceptualization:** Junya Zhang, Robert C. Augustine, Richard D. Vierstra.

**Formal analysis:** Junya Zhang.

**Funding acquisition:** Masaharu Suzuki, Donald R. McCarty, Richard D. Vierstra.

**Investigation:** Junya Zhang, Robert C. Augustine, Juanjuan Feng, Si Nian Char.

**Methodology:** Junya Zhang.

**Resources:** Masaharu Suzuki, Si Nian Char, Bing Yang, Donald R. McCarty.

**Supervision:** Richard D. Vierstra.

**Writing – original draft:** Junya Zhang, Robert C. Augustine.

**Writing – review & editing:** Junya Zhang, Richard D. Vierstra.

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
