## [Decision Letter · Decision Letter 0]

5 Aug 2021

Dear Dr Vierstra,

Thank you very much for submitting your Research Article entitled 'The SUMO Ligase MMS21 Profoundly Influences Maize Development Through its Impact on Genome Activity and Stability' to PLOS Genetics.

The manuscript was fully evaluated at the editorial level and by independent peer reviewers. The reviewers appreciated the attention to an important topic but identified some concerns that we ask you address in a revised manuscript.

As you will see from the detailed comments of the reviewers, they appreciate the novel insight into the role of the MMS21 protein generated by the analysis of a mutant series in maize. The results indicate similar, but also important differences compared to Arabidopsis data, and the reviewers see this as a solid study that provides several aspects for future work. Please improve the Discussion about the comparison between maize and Arabidopsis as suggested by reviewer 1, answer the comments on the degree of endoreplication and ploidy, and make sure all references appear in the correct order and context.

We ask you to modify the manuscript according to the review recommendations. Your revisions should address the specific points made by each reviewer.

[LINK]

Yours sincerely,

Ortrun Mittelsten Scheid

Associate Editor

PLOS Genetics

Claudia Köhler

Section Editor: Plant Genetics

PLOS Genetics

Reviewer's Responses to Questions

**Comments to the Authors:**

Reviewer #1: This is review of the manuscript „The SUMO ligase MMS21 profoundly influences maize development through its impact on genome activity and stability“ submitted by Zhang et al. to the PLoS Genetics. This manuscript represents the first genetic and biochemical analysis of E3 SUMO ligase MMS21 functions in maize. Using different methods, the authors have isolated a series of MMS21 loss-of-function mutant alleles that were characterized as to the whole plant, inflorescence and kernel phenotypes. In general, the plants carrying weaker mms21 alleles were smaller, had reduced fertility and showed reduced kernel weight. The strong mutant alleles caused seed or early seedling lethality. The transcriptomic analysis revealed a typical SMC5/6 complex mutant expression pattern with upregulated DNA damage repair genes. Surprisingly, there was also a transcriptional upregulation of Zein genes in the somatic tissues. However, Zein mRNA was not translated into the protein. Another interesting observation was that some of the upregulated genes occurred in clusters, suggesting a possible role of MMS21 (or the whole SMC5/6 complex) in transcriptional repression. The mms21 mutants were also hypersensitive to several DNA damaging agents, suggesting that MMS21 is involved in multiple DNA damage repair pathways. Via biochemical analyses, the authors show that MMS21 interacts with SCE1 SUMO conjugating enzyme and two subunits of SMC5/6 complex, namely SMC5 and NSE4a.

This is a well conducted study that provides a plethora of information on the role of MMS21 in maize.

Specific comments:

Introduction. The MMS21 protein is a known and evolutionary conserved subunit of the SMC5/6 complex. This is somehow lost in the Introduction and I suggest making this point more prominent. Furthermore, it would be great mentioning that MMS21 / HPY2 is known also as NSE2.

The citations in the text often do not follow a chronological order. Please correct throughout the manuscript.

The authors analyzed endoreduplication levels using seeds of mms21 plants and did not observe any difference to WT. Is it possible that the maize seeds already reached their maximum endoreduplication capacity and therefore do not show an enhanced phenotype in the mutants? All quantifications of endoreduplication levels in Arabidopsis (which the authors refer to) were done using somatic tissues. Therefore, I suggest repeating the experiment using leaf tissues.

The authors suggest defects in anther development, pollent germination and problematic seed development. This is reminiscent of a recently published results in Arabidopsis showing that mms21/nse2 mutants produce unreduced gametes and triploid progeny (Yang et al., Plant Cell, 2021). It would be great if the authors could analyze ploidy of the maize mms21 plants (in particular the poorly growing ones).

Line 522. I cannot fully agree with the sentence „Strikingly, while strong Arabidopsis MMS21 mutants are viable and fertile (Huang et al., 2009; Ishida et al., 2009), we found that the strongest maize alleles impacting MMS21 not only substantially attenuated fertility and seed formation, but also compromised vegetative development leading to progeny that either failed to germinate or arrested growth soon after germination“. Strong Arabidopsis mms21 mutants are also affected in their growth and fertility (e.g. Liu et al., BMC Plant Biol., 2014; Yang et al., Plant Cell, 2021). In fact, the phenotypes are very similar to what the authors describe for maize. Therefore, I suggest rewriting the sentence starting on line 522 and reformulating the sentence starting on line 526.

Reviewer #2: The manuscript “The SUMO ligase MMS21 profoundly influences maize development through impact on genome activity and stability” by Zhang et al. is a thorough assessment of the role of a SUMO ligase in cellular physiology.

The authors find wide mis-expression of genes if SUMO ligase MMS21 is compromised by a mutation. The data indicate moreover that complete loss of function alleles are lethal, uncovering one node of essentiality of the SUMO conjugation system in higher plants.

The most significant asset of this manuscript is the thorough data collection and analysis. This is the best possible basis to assess functions of the MMS21 protein, and to compare this protein´s importance across species.

Just to give an example, the authors found that MMS21 of maize is not closely involved in the endoreplication processes (higher ploidy levels are occurring regularly in large plant cells such as trichomes), whereas the previous data in Arabidopsis had indicated such a function. This manuscript shows that endoreplication is not a process affected across the plant kingdom. In contrast, work with the large genome species maize once more emphasized the importance of MMS21 to regulate gene expression, presumably at the level of hetero- versus euchromatin.

The manuscript offers a number of novel aspects for further in-depth analysis. For instance, a group of storage proteins is mis-regulated (ectopically induced), but the proteins do not accumulate, indicating further regulatory steps post transcription. Finally, the significant number of mutants generated, combined with in vitro activity assays, allowed to pinpoint the amino terminus as an important region for protein activity. These mutants can be used to investigate in vivo complexes containing MMS21 and their localization on chromatin in the future.

**Have all data underlying the figures and results presented in the manuscript been provided?**

Reviewer #1: Yes

Reviewer #2: Yes

PLOS authors have the option to publish the peer review history of their article (what does this mean?). If published, this will include your full peer review and any attached files.

Reviewer #1: No

Reviewer #2: No

---

## [Decision Letter · Decision Letter 1]

20 Sep 2021

Dear Dr Vierstra,

We are pleased to inform you that your manuscript entitled "The SUMO Ligase MMS21 Profoundly Influences Maize Development Through its Impact on Genome Activity and Stability" has been editorially accepted for publication in PLOS Genetics. Congratulations!

Yours sincerely,

Ortrun Mittelsten Scheid

Associate Editor

PLOS Genetics

Claudia Köhler

Section Editor: Plant Genetics

PLOS Genetics

Comments from the reviewers (if applicable):

Reviewer's Responses to Questions

**Comments to the Authors:**

Reviewer #1: The authors have successfully addressed all my concerns.

**Have all data underlying the figures and results presented in the manuscript been provided?**

Reviewer #1: Yes

PLOS authors have the option to publish the peer review history of their article (what does this mean?). If published, this will include your full peer review and any attached files.

Reviewer #1: **Yes: **Ales Pecinka

**Data Deposition**

http://datadryad.org/submit?journalID=pgenetics&manu=PGENETICS-D-21-00956R1

**Press Queries**

---

## [Editor Report · Acceptance letter]

11 Oct 2021

PGENETICS-D-21-00956R1 

The SUMO Ligase MMS21 Profoundly Influences Maize Development Through its Impact on Genome Activity and Stability 

Dear Dr Vierstra, 

We are pleased to inform you that your manuscript entitled "The SUMO Ligase MMS21 Profoundly Influences Maize Development Through its Impact on Genome Activity and Stability" has been formally accepted for publication in PLOS Genetics! Your manuscript is now with our production department and you will be notified of the publication date in due course.

With kind regards,

Anita Estes

PLOS Genetics

On behalf of:
